# Direct observation of two-channel photodissociation of carbon monoxide from the hemoglobin subunits

Sergei V. Lepeshkevich [1] ✉, Igor V. Sazanovich [2] ✉, Marina V. Parkhats [1], Syargey N. Gilevich [3], Aliaksei V. Yantsevich[3], Julia A. Weinstein [4], Michael Towrie [2] & Boris M. Dzhagarov[1]

Determining dynamics of bond breaking between carbon monoxide (CO) and heme proteins is essential to understand the interplay between protein function and dynamics, which is one of the fundamental challenges of physical biology. There is an ongoing debate about the mechanism of CO photodissociation from the heme iron. Here we use picosecond to millisecond transient mid-infrared spectroscopy to determine the dynamics of CO photodissociation from the isolated human hemoglobin chains. We find that the breaking of the Fe–CO bond is not a single-step process as is commonly accepted, but rather at least a two-step process, which includes both the known prompt sub-50-fs CO dissociation event and the additional slower, ~15 ps CO dissociation process discovered in this study. These results offer the direct experimental proof of the CO photodissociation mechanism containing several dissociative states.

Bond breaking and formation between small ligands and heme proteins is a trigger for many important chemical and biological processes ranging from protein conformational changes[1–3], electron transfer[4], to cellular signaling[5]. These processes play a fundamental role in chemistry, biology, and medicine. A model for studying such processes in multisubunit proteins, is hemoglobin (Hb), which carries molecular oxygen ($O_2$) in erythrocytes from lungs to tissues[6]. Human Hb is a tetramer composed of two α and two β subunits (Fig. 1a)[7]. Each subunit contains one identical heme group, in which the iron ion is bound to the proximal histidine (His) (Fig. 1c, d). The ferrous heme iron can reversibly bind one diatomic ligand, such as $O_2$ or carbon monoxide (CO), on the opposite side of the proximal His (Fig. 1c, d), in the distal heme pocket[6]. CO bound to the heme iron is oriented almost perpendicular to the heme plane[8,9]. The CO ligand, being produced endogenously by heme oxygenases, is a signaling molecule in several pathways, which involve the CO binding to ferrous heme proteins[5]. Thus, the study of the dynamics of the bond breaking and formation

between the CO ligand and the heme attached to Hb is essential for insight into the primary processes in cellular signaling and small molecules transport in physiological systems.

One way to break the bond between the ferrous heme iron and CO is by photolysis[10]. The quantum yield of CO photodissociation for carbonmonoxy hemoglobin (HbCO) after photoexcitation into the Q bands of the heme moiety (Fig. 1b) is considered to be unity[11–14]. In less than 1 ps after the photodissociation[15], the CO molecule translocates from its binding site (Fig. 1c, d) to a primary docking site located above the heme group within a few angstroms[16] from the heme iron. This translocation of CO involves ligand translation as well as rotation. The primary docking site constrains CO to lie approximately parallel to the plane of the heme[8], an orientation approximately orthogonal to that of the heme-bound CO. In the α and β subunits within human Hb at ambient temperature, the CO molecule remains trapped in the primary docking site at most a few nanoseconds[16]. Subsequently, the CO molecule can rebind to the heme iron or migrate through the protein

[1]B.I. Stepanov Institute of Physics, National Academy of Sciences of Belarus, Minsk, Belarus. [2]Central Laser Facility, Research Complex at Harwell, STFC Rutherford Appleton Laboratory, Harwell Campus, UK. [3]Institute of Bioorganic Chemistry, National Academy of Sciences of Belarus, Minsk, Belarus. [4]University of Sheffield, Department of Chemistry, Dainton Building, Brook Hill, Sheffield, UK. ✉e-mail: s.lepeshkevich@ifanbel.bas-net.by; igor.sazanovich@stfc.ac.uk

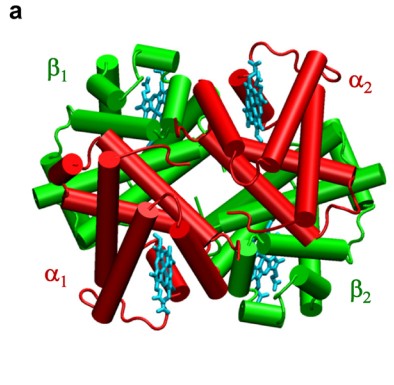

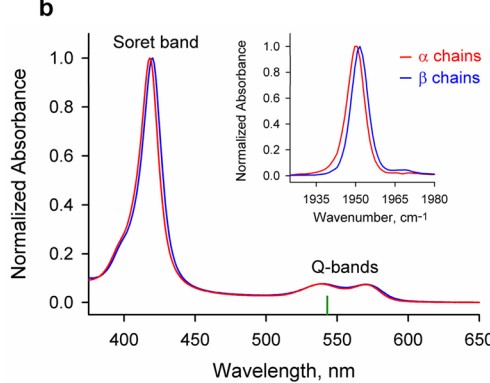

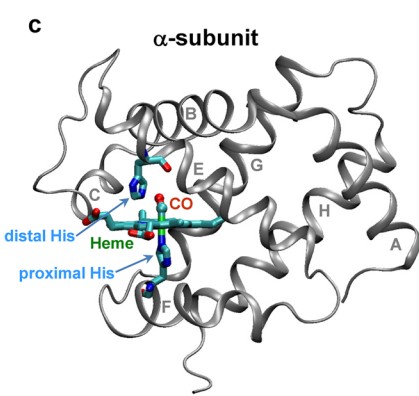

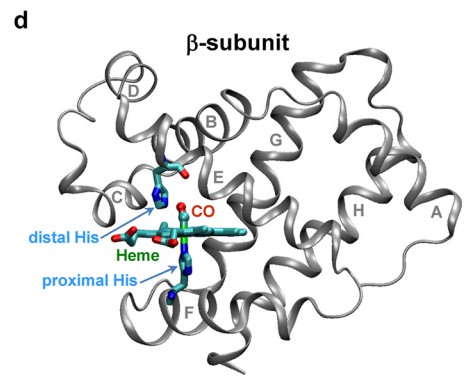

**Fig. 1 | Structures and absorption properties of the α and β subunits of human HbCO. a** The quaternary arrangement of the α and β subunits of HbCO (PDB entry 1BBB)[7]. The α and β subunits in red and green, respectively, are shown with the hemes in cyan. **b** Visible (the main figure) and mid-infrared absorption spectra (the inset) of the isolated carbonmonoxy α and β chains. The absorption spectra for the α and β chains are shown by the red and blue lines, respectively. The absorbance in the visible region is due to the heme group present in the proteins. The excitation wavelength, $\lambda_{exc} = 543$ nm, is indicated as the vertical green mark. In the inset, the FTIR spectra of the stretching bands of the heme-bound CO are shown. The visible absorption spectra for the α chains were repeated 7 times, and 8 times for the β chains. The mid-infrared absorption spectra were repeated 8 times for each sample. The structure of the α subunit (**c**) and the β subunit (**d**) of human HbCO (PDB entry 1BBB[7]). The heme group, the distal histidine as well as the proximal histidine, which links the heme iron to the F helix, are labeled and shown in sticks. The CO molecule bound to the heme iron in the distal heme pocket is labeled in red. Location of the CO binding site is marked by the heme-bound CO. Source data are provided as a Source Data file.

matrix into the solvent, from where it can return into the protein. The CO rebinding from within the protein matrix (the geminate CO rebinding) occurs on a nanosecond timescale at ambient temperature[17–19], while the CO rebinding from the solvent (the bimolecular CO rebinding) occurs on a microsecond to millisecond timescale[20–22]. Taking advantage of the photosensitivity of the chemical bond between the heme iron and CO, nanosecond and picosecond time-resolved transient absorption and resonance Raman spectroscopy have been used for kinetic studies of the ligand rebinding[17–23] and tertiary/quaternary conformational changes in the heme proteins[6,24–29] following the ligand photodissociation. Using the femtosecond transient absorption spectroscopy, it has been shown that the photolysis occurs in less than 50 fs[30]. Further research by ultrafast transient absorption as well as resonance Raman spectroscopy was directed mainly towards investigating energy relaxation[12,30–33] and primary conformational changes in the heme proteins[34] after the ligand dissociation.

Many experimental and theoretical works have been devoted to study the mechanism of photodissociation of carbonmonoxy-heme proteins as well as the electronic[35,36] and structural relaxation[3,36–38] that accompanies and follows the photodissociation. There is still an ongoing debate in the literature with respect to the dissociative state(s) of carbonmonoxy-heme proteins[35,36,39–43]. Different models have been suggested for describing the relevant excited states[12,35,39]. It should be stressed that until now, there has been no experimental evidence for the existence of a dissociative channel other than the

prompt sub-50-fs one. Any additional dissociative states could be expected to be relatively long-lived with smaller contributions to the overall dissociation process. An observation of the photodissociation from such states would be significantly hindered by signals from the primary sub-50-fs photolysis. However, this experimental hurdle can be overcome by direct detection of the CO ligand vibrational signal (rather than those of the heme prosthetic group) in the mid-infrared (mid-IR) region during and after the course of the photodissociation. The mid-IR spectrum of HbCO shows discrete CO stretch bands[44], denoted $A_0$ (-1968 cm$^{-1}$) and $A_1$ (-1950 cm$^{-1}$)[45], well separated from the main vibrational bands of the heme protein itself. These peaks reflect distinct conformational substates of the ground-state HbCO. The CO stretch bands for the heme-bound CO are susceptible to changes in the iron-ligand bond and the electric field at the active site[46]. The existence of CO stretch bands in the mid-IR region allows direct observation of the bound CO ligand[8,47,48], monitoring of changes in its coordination to the heme iron[49–51], and tracking of the heme−CO bond dissociation dynamics[52] by time-resolved mid-IR spectroscopy. Furthermore, the transient IR spectroscopy allows direct monitoring of the photolyzed CO and its diffusion inside the protein matrix[15,53]. For human Hb, the distinct stretch bands for CO photolyzed and temporarily trapped in the protein matrix are detected in the region of 2090–2160 cm$^{-1}$ [9], close to the stretching frequency for free CO molecule (-2143 cm$^{-1}$)[54]. The spectral positions and absorption intensities of the vibrational bands for the photolyzed CO are sensitive to ligand relocation to docking sites within the protein, rotational motions of the ligand in

these sites, and protein conformational changes[46], with the time-resolved spectra of photolyzed CO being mainly determined by the interaction between the CO dipole and the surrounding electric field. Additionally, the relative orientation of CO bound to and dissociated from the heme iron can be determined with time-resolved mid-IR polarization spectroscopy[8,9,15,55–57].

In the present work, using a picosecond to millisecond time-resolved mid-IR spectrometer, we discover the CO photodissociation from the long-lived excited state(s) of HbCO and resolve the mechanism of the heme–CO photodissociation. We discover that the breaking of the chemical bond between the ferrous heme iron and CO involves not only the previously known prompt sub-50-fs CO dissociation process but also the slower (~15 ps) CO dissociation process. Direct observation of the two-step photodissociation of CO in the present work is the experimental proof of the existence of several dissociative states of HbCO.

## Results

### Time-resolved transient mid-IR absorption spectra

As models for the α and β subunits within HbCO, we chose the isolated carbonmonoxy α and β chains, respectively, which are frequently used

to solve the problem of functional differences between the α and β subunits within human HbCO[20]. The study of the isolated chains avoids ambiguity in the interpretation of experimental data for the native tetrameric Hb consisting of two different types of subunits. The isolated chains exhibit significant non-equivalence in ligand rebinding[58,59]. Besides, in the α and β chains, photophysical processes resulting in the ligand photodissociation are expected to occur in the same electronically excited states of the heme–ligand complex. Therefore, in the present work, the data obtained separately for the isolated α and β chains were combined and analyzed together to aid assignment of the spectral features to reaction intermediates.

Picosecond to millisecond time-resolved transient absorption spectra in the mid-IR region were measured on the ULTRA[60] apparatus in the Time-Resolved Multiple Probe Spectroscopy mode[61] (for details see the "Methods" section). The CO complexes of the isolated Hb chains were photoexcited into the Q band at 543 nm (Fig. 1b). The transient mid-IR absorption spectra were detected in the time range from 2 ps up to 800 μs in the spectral region of 1880–2005 cm$^{-1}$ (Fig. 2), where the heme-bound CO absorbs (Fig. 1b, inset). Additionally, on the same timescale, the transient vibrational spectra were measured in the spectral region of 2005–2160 cm$^{-1}$, where the

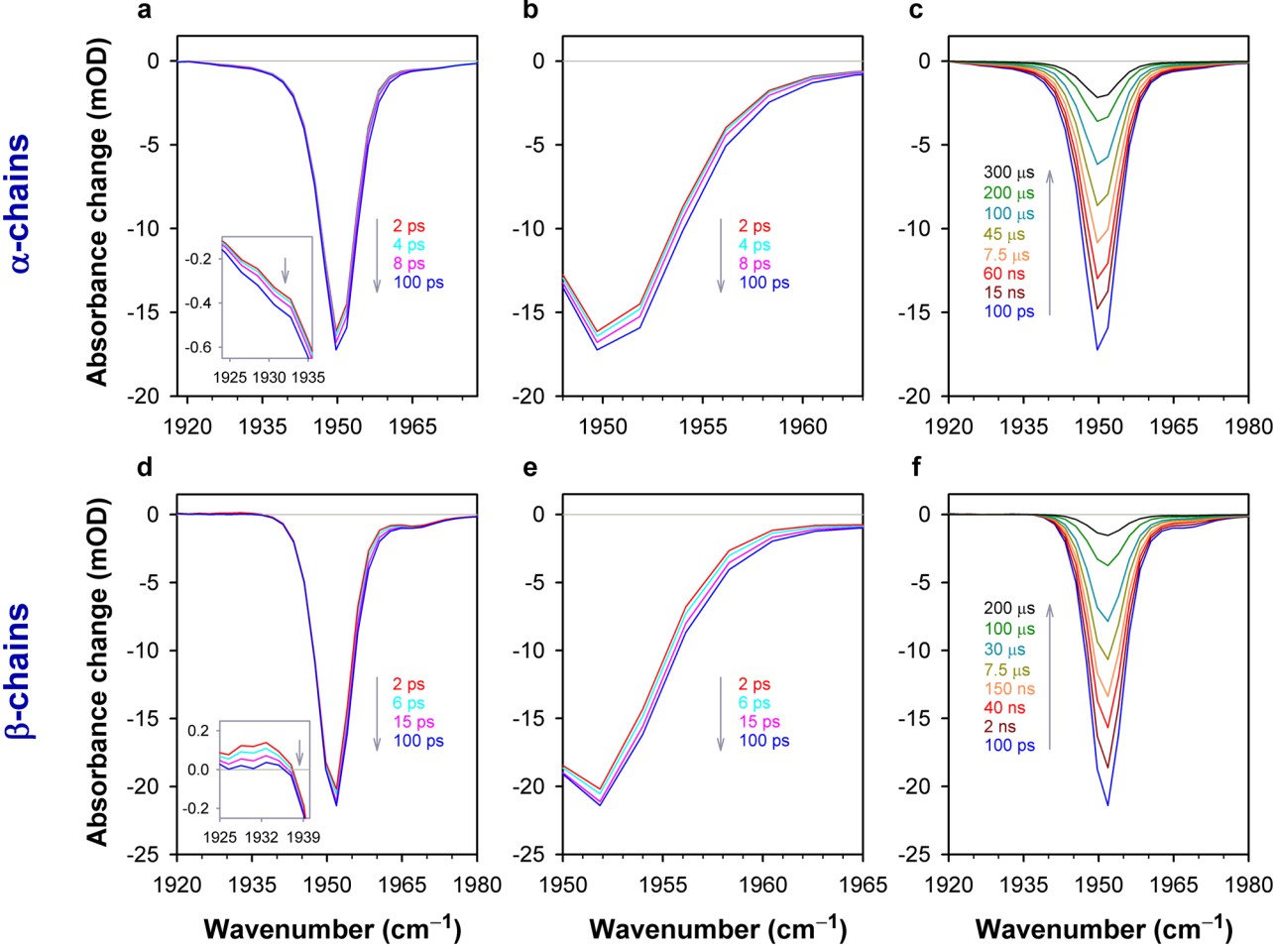

**Fig. 2 | Time-resolved transient mid-IR absorption changes of the isolated carbonmonoxy chains of human HbCO in the region of ground-state bleach.** **a**, **d** The representative spectra detected on the 2–100 ps timescale after the photoexcitation. **b**, **e** Zoomed-in spectra shown in (**a**) and (**d**), respectively, in the high-frequency region of the negative band centered at ~1950 cm$^{-1}$. **c**, **f** The representative spectra detected in the time range from 100 ps to hundreds of microseconds. The spectra were obtained for the isolated carbonmonoxy α chains (**a**–**c**) and carbonmonoxy β chains (**d**–**f**). The arrows indicate the time evolution of the

signal. The spectra shown correspond to the magic angle relative polarization between the pump and the probe beams. The time-resolved mid-IR measurements were repeated 4 times for each sample. In a single measurement, typically 75 time delay points were used. Each time-resolved mid-IR spectrum shown corresponds to 150,000 averaged laser shots. Conditions: 50 mM Tris buffer, pD 8.2, at 19 °C. Concentrations of the carbonmonoxy α and β chains were 3.0 and 4.0 mM in heme, respectively. Excitation wavelength, $\lambda_{exc}$ = 543 nm. Source data are provided as a Source Data file.

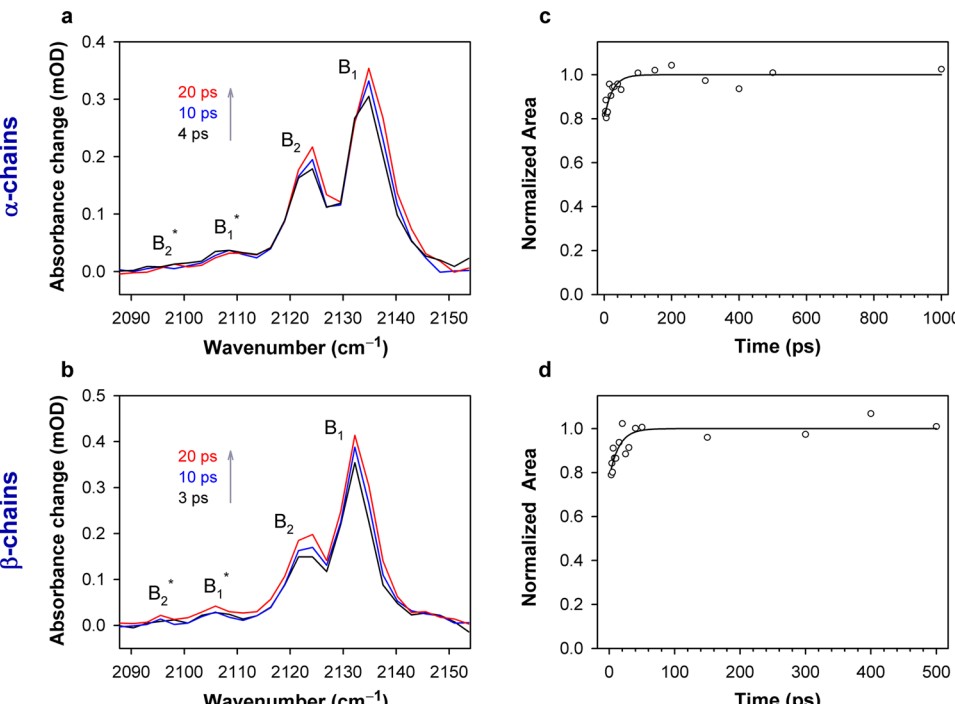

**Fig. 3 | Representative time-resolved vibrational spectra obtained on the picosecond timescale for CO photolyzed from the isolated carbonmonoxy chains of human HbCO.** The transient spectra obtained for the isolated carbonmonoxy α chains (**a**) and carbonmonoxy β chains (**b**) at selected time delays (indicated in the key). For clarity, the background has been subtracted from the measured spectra. The arrows indicate the time evolution of the signal. The spectra shown correspond to the magic angle relative polarization between the pump and the probe beams. Similar spectra were obtained for the other two (parallel and perpendicular) polarization conditions. Time dependence of the normalized total integrated area of the vibrational spectra of CO photolyzed from the isolated carbonmonoxy α chains (**c**) and carbonmonoxy β chains (**d**). To calculate the total integrated area, the transient bands corresponding to the CO molecules, produced in both the ground vibrational state (v = 0) and the first excited vibrational state (v = 1), were taken into account. The total integrated area was calculated for the spectral region of 2090 to 2160 cm$^{-1}$ after the baseline subtraction. The normalized areas at selected time delays after the photoexcitation are shown as open circles. The time delays, *t*, range from 3 to 1000 ps (**c**) and from 3 to 500 ps (**d**). The solid line is a fit to the equation: $A(t) = A_{max} \cdot (1 - A_{inc} \cdot \exp(-t/\tau_{inc}))$, where $A_{max}$ is the maximum area value, $A_{inc}$ is the increase in the normalized area, $\tau_{inc}$ is the time constant for the increase in the normalized area. The time constant $\tau_{inc}$ was found to be ~20 ps, the increase in the normalized area being no more than 0.2. The time-resolved mid-IR measurements were repeated 4 times for each sample. In a single measurement, typically 75 time delay points were used. Each time-resolved mid-IR spectrum shown on (**a**, **b**) or each time delay point shown on (**c**, **d**) corresponds to 150,000 averaged laser shots. Conditions: 50 mM Tris buffer, pD 8.2, at 19 °C. Concentrations of the carbonmonoxy α and β chains were 3.0 and 4.0 mM in heme, respectively. Excitation wavelength, λ$_{exc}$ = 543 nm. Source data are provided as a Source Data file.

photolyzed CO absorbs (see, for example, Fig. 3a, b). In order to elucidate the orientation of the CO ligand with respect to the heme plane in the bound and photolyzed states, the data were recorded at three different configurations of the relative polarization, *P*, of the pump and probe beams: parallel (0°), perpendicular (90°), and the "magic" angle (54.7°) configurations. Determination of the CO orientation by photoselection spectroscopy is possible because spectroscopic transitions for both heme and CO are polarized along specific directions in the molecular frame. The representative time-resolved transient mid-IR absorption spectra shown in Figs. 2 and 3a, b were measured at the "magic" angle relative polarization, at which the measured spectra are not affected by molecular orientation. Similar time-resolved spectra were obtained for the other two (parallel and perpendicular) polarization conditions (see, for example, Supplementary Fig. 1). The measured spectra depend on both the type of the Hb chains as well as the polarization.

The time-resolved vibrational spectra for CO photolyzed from the isolated carbonmonoxy α and β chains are shown in Fig. 3a and 3b, respectively. The photolyzed CO absorbs at frequencies between 2090 and 2150 cm$^{-1}$. Two major positive transient bands labeled B$_1$ and B$_2$ (Fig. 3a, b) correspond to the photolyzed CO molecules produced in the ground vibrational state (v = 0) and oriented oppositely in the primary docking site[8,62]. For the α chains, the B$_1$ and B$_2$ bands are at 2136 and 2124 cm$^{-1}$, while for the β chains those are at 2133 and

2123 cm$^{-1}$ (Fig. 3a, b), respectively. The splitting between the maxima of the B$_1$ and B$_2$ bands observed for the α and β chains was found to be ~12 and ~10 cm$^{-1}$, respectively, agreeing well with that of 10.5 cm$^{-1}$ published for HbCO[63]. Two satellite features labeled B$_1$* and B$_2$* (Fig. 3a, b) arise from the CO generated in the first excited vibrational state (v = 1)[64]. (In Fig. 3a, the B$_2$* band is not well resolved due to signal-to-noise limitation.) The B$_1$* and B$_2$* bands is a copy of two major B$_1$ and B$_2$ bands with lower amplitude and red-shifted due to the anharmonicity of the vibrational ladder of CO[63]. The population of CO in its first excited vibrational state (v = 1) was found to be about 5%. This value was obtained within the harmonic oscillator approximation, where the integrated absorbance for a vibrational transition is proportional to (v + 1)[65]. Here, the absorbance cross section for the first hot band transition (1→2) was assumed to be twice of that for the ground state transition (0→1). Therefore, under our experimental conditions, photolysis of the heme proteins produces a small population of CO in its first excited vibrational state. Moreover, the photolyzed CO is not being formed in any vibrational states higher than v = 1. The decay of the first excited vibrational state (v = 1) of the photolyzed CO trapped in the primary docking site is known to occur over hundreds of picosecond[64].

The initial anisotropy of the vibrational spectra for photolyzed CO was used to determine the orientation of unbound CO located in the primary docking site (for details see Supplementary Note 2). Assuming

that the IR transition dipole of the CO molecule is along the CO axis, the average angle Θ between the CO bond and the heme plane normal in both Hb chains was found to be 69° ± 6° for both $B_1$ and $B_2$ photoproduct states. The obtained results indicate that, in the primary docking site of both the α and β chains, CO lies approximately in the plane of the heme (as opposed to the heme-bound CO.)

The time-resolved vibrational spectra in Fig. 3a, b change noticeably with time. In particular, the total integrated area of all the transient bands corresponding to the photolyzed CO, produced in both the ground and first excited vibrational states, was found to increase up to 20 % with a time constant of ~20 ps (Fig. 3c, d). It should be noted that the observed increase in the total integrated area cannot be due to the vibrational relaxation of the photolyzed CO trapped inside the protein matrix, since the last process is much slower (hundreds of picoseconds)[64]. The observed changes in the vibrational spectra can be induced by the actual increase in the population of the photodissociated CO on the ~20 ps timescale.

The transient mid-IR absorption spectra observed in the region of 1880–2005 cm$^{-1}$ (Fig. 2) are dominated by the negative band at ~1950 cm$^{-1}$, which corresponds to the stretching frequency of CO bound to the Hb chains in the $A_1$ conformational substate[45]. The performed measurements exposed a series of time-dependent changes in the transient IR absorption spectra. On the 2–100 ps timescale after the photoexcitation, changes in the shape of the negative band centered at ~1950 cm$^{-1}$ are observed for both the α chains (Fig. 2a, b) and the β chains (Fig. 2d, e). Much later, on the timescale from 100 ps to 800 μs, this negative band is seen to decay completely (Fig. 2c, f).

Using singular value decomposition (SVD)[66] and maximum entropy method (MEM) analysis[67], we represented the measured transient IR absorption spectra as the sum of the spectra associated with the intermediate species evolving in the first 100 picoseconds after the photoexcitation (Fig. 4a, b) and those evolving over a much longer timescale ranging from nanoseconds to hundreds of microseconds (Fig. 4c, d). For details on SVD, MEM analysis, and the procedure for calculating the associated spectra we refer the reader to Supplementary Note 1 & Supplementary Tables 1 and 2. Subsequently, the spectra associated with the fast and slow-evolving species were analyzed separately. To provide a compact representation of the entire set of the analyzed spectra at the three polarization settings, the corresponding spectra were subjected to the global SVD analysis. In the obtained representation, the time-dependent amplitudes for the three polarization settings refer to the same set of basis spectra. The SVD components that provide the best least-squared approximation of the analyzed spectra for the α and β chains are shown in Figs. 5 and 6, respectively, and are discussed in the next sections.

### Geminate and bimolecular CO rebinding

For the α chains in the spectral region of 1880–2005 cm$^{-1}$, the transient IR absorption spectra associated with the slow evolving species are described at the three polarization settings by only one basis spectrum, $U_1$ (Fig. 5b, inset). For the β chains in the corresponding region, two basis spectra, $U_1$ and $U_2$, are required (Fig. 6b, c, insets), $U_1$ making the dominant contribution (~99%). As it is seen from the insets in Figs. 5b and 6b, $U_1$ is identical (up to a negative normalization

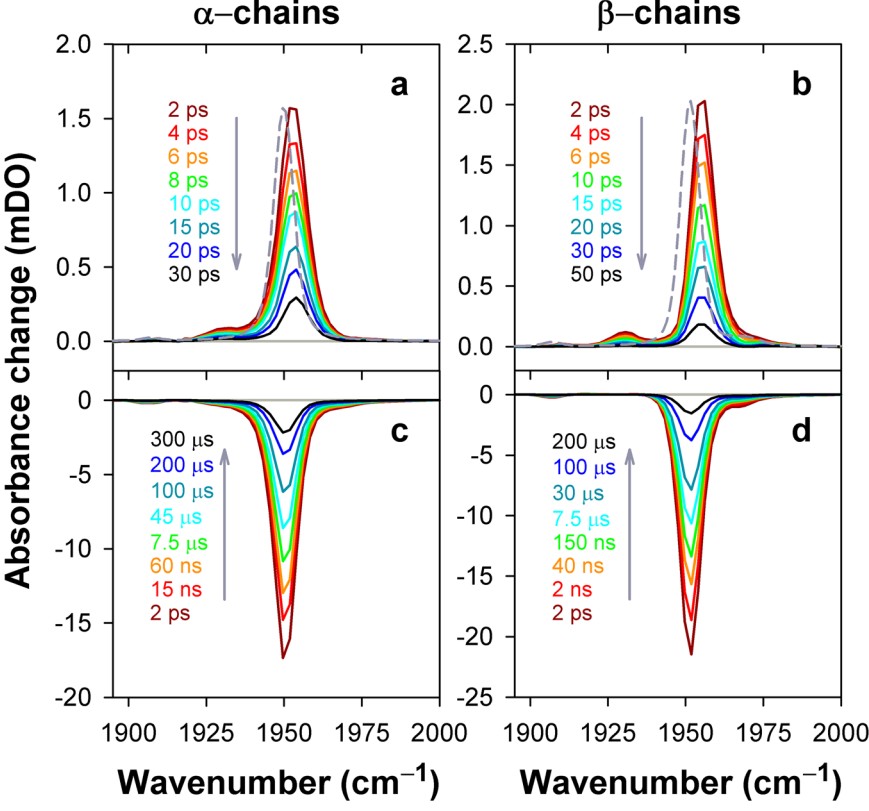

**Fig. 4 | Photoinduced spectral changes associated with the fast (picoseconds) and slow (nanoseconds to microseconds) evolving species after the photoexcitation of the isolated carbonmonoxy chains of human HbCO. a, b** The representative transient mid-IR absorption spectra associated with the intermediate species evolving in the first 100 picoseconds after the photoexcitation (solid lines). The ground-state FTIR absorption spectrum normalized to an arbitrary value is shown as the short-dash line for comparison. **c, d** The representative transient mid-IR spectra associated with the intermediate species evolving over a timescale ranging from nanoseconds to hundreds of microseconds. The transient IR spectra were obtained for the isolated α chains (**a, c**) and the isolated β chains (**b, d**). The pump-probe delays are color-coded in the transient spectra. The arrows indicate the time evolution of the signal. The shown transient spectra correspond to the magic angle relative polarization between the pump and the probe beams. Similar spectra were obtained for the other two (parallel and perpendicular) polarization conditions. Note the difference in the time range for the data shown in (**a, b**) vs. (**c, d**). Source data are provided as a Source Data file.

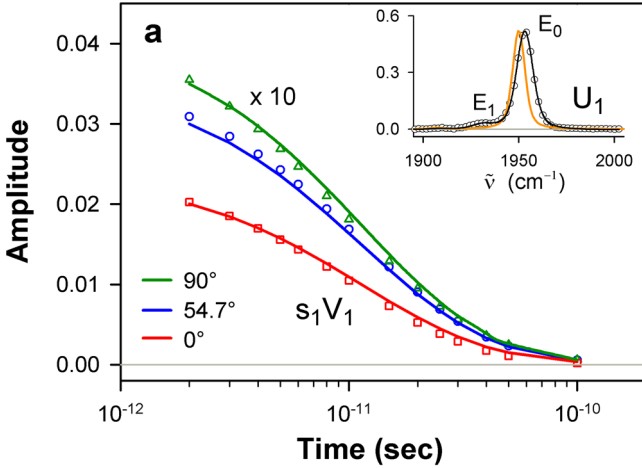

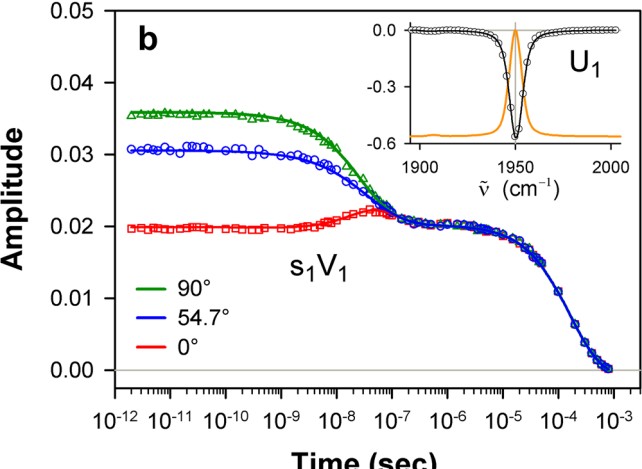

**Fig. 5 | Global analysis of the photoinduced mid-IR absorption changes of the isolated carbonmonoxy α chains in the region of ground-state bleach at the parallel, perpendicular, and magic angle polarization settings. a** Global SVD of the spectra associated with the intermediate species evolving in the first 100 picoseconds after the photoexcitation. **b** Global SVD of the spectra associated with the intermediate species evolving over a timescale ranging from nanoseconds to hundreds of microseconds. Global SVD in (**a**), as in (**b**), was performed on the global data matrix composed of data obtained at three polarization settings. In (**a**), as in (**b**), the time-dependent amplitudes ($V_1$) of the first basis spectrum, multiplied by the first singular value ($s_1$), at three polarization settings were globally fitted to eqn (S5) with the lifetime distribution $g(\log\tau)$ being the same at all these polarization settings, the initial anisotropy $r_0$ (eqn (S6)) being the global parameter (see Supplementary Note 2 for details). Additionally, at the global fitting in (**b**), the rotational correlation time $\tau_{rot}$ (eqn (S6)) was taken as a global parameter. In (**a**) and (**b**), the time-dependent amplitudes $s_1V_1$ and their global fit obtained with the simultaneous analysis of the data under all the polarization settings are plotted as symbols and the solid lines, respectively. The data obtained at three polarization settings are color-coded: 0° (red), 54.7° (blue), and 90° (green). In all insets, the first basis spectrum ($U_1$) and its fit with a sum of Voigt functions are shown in black as circles and the solid line, respectively. The ground-state FTIR absorption spectrum normalized to an arbitrary value is shown as the orange solid line for comparison. In the inset of (**b**), the ground-state FTIR absorption spectrum is additionally offset for clarity. The vertical gray line shows the position of the vibrational band of CO in the $A_1$ conformational substate. Note the difference in the range for the horizontal axis in (**a**) and (**b**). Source data are provided as a Source Data file.

constant) to the ground-state FTIR absorption spectrum, the positions and widths of the corresponding bands being the same within the experimental accuracy (Supplementary Table 3). For each Hb chain, the first basis spectrum, $U_1$, represents the heme-bound CO species that have been removed from the ground state due to the

photoexcitation, and transferred to some other state(s) (as explained below). The first basis spectrum, $U_1$, caused by the depopulation of the ground state represents the bleaching of the ground-state IR absorption (commonly termed as ground-state bleach). $U_1$ exhibits a prominent band at ~1950 cm⁻¹ (Supplementary Table 3), which corresponds to the stretching frequency of CO bound to the Hb chains in the $A_1$ conformational substate[45]. The first amplitude vector $s_1V_1$ (Figs. 5b and 6b) at the magic angle polarization setting represents a very good approximation to the kinetics of CO rebinding to the isolated chains in the $A_1$ substate. For each Hb chain, this $s_1V_1$ vector, containing time-dependent amplitudes, was subjected to the MEM analysis which derived one model-independent log-lifetime distribution, $g(\log\tau)$, from the data. The fit $F(t)$ to the amplitude vector $s_1V_1$ at time $t$ can be written as

$$F(t) = F_0 \int_{-\infty}^{+\infty} g(\log\tau)e^{-t/\tau} d\log\tau \qquad (1)$$

where $F_0$ is a normalization constant, $g(\log\tau)$ is the lifetime distribution that corresponds to decaying kinetics. The extracted lifetime distributions for the α and β chains are shown in Fig. 7 by black and blue lines, respectively. The bands that appear in the lifetime distribution between 10⁻¹⁰ and 10⁻⁶ s are associated with the geminate CO rebinding, while those between 10⁻⁵ and 10⁻³ s are associated with the bimolecular CO rebinding. Significant functional non-equivalence of the α and β chains in the geminate CO rebinding is obvious from the data. The α chains show only one geminate rebinding phase, while the β chains show two distinct geminate rebinding phases (both prompt and delayed ones) with similar fractional contributions (Supplementary Table 4, $F_1$ and $F_2$). The prompt geminate CO rebinding phase in the β chains was found to be faster than the one in the α chains (Supplementary Table 4). The observed α/β difference in the geminate CO rebinding agrees well with that reported previously[16] for the α and β subunits within human Hb and confirms the validity of using the isolated chains as models for the corresponding subunits within tetrameric hemoglobin. As revealed in the earlier study[16] by time-resolved Laue crystallography of photolyzed HbCO, there is a correlation between the rate constant for the CO rebinding from the primary docking site within the distal heme pocket (Fig. 1c, d) and the CO center-of-mass displacement during this process. The distance between the CO binding site and the primary CO docking site in the β subunits has been found to be ~0.25 Å smaller than that in the α subunits (~1.83 Å at 100 ps), and the CO rebinding from the primary docking site is faster in the β subunits, suggesting distal control of the CO rebinding dynamics.

It should be noted that the ground state bleach signal due to the ligand dissociation is spectrally separated from the photolyzed CO absorption signal by about 180 cm⁻¹ and differs by more than an order of magnitude in absorbance. For the isolated carbonmonoxy α and β chains, the integrated area of the bleach signal was found to be 36 ± 3 times larger than that of the photolyzed CO absorption signal at the magic angle polarization setting at a delay of 100 ps, agreeing well with the data for tetrameric Hb¹³CO[62]. The measured anisotropy of the ground-state bleach spectra was used to determine the geometry of the bound CO in the Hb chains (for details see Supplementary Note 2). The average angle Θ between the CO bond and the heme plane normal in both Hb chains was found to be 17 ± 1° (Supplementary Table 5). The obtained results indicate that the isolated chains are mainly in the $A_1$ conformational substate with the coordinated CO molecule almost perpendicular to the heme plane.

## Long-lived dissociative excited state

For each Hb chain in the spectral region of 1880−2005 cm⁻¹, the transient IR absorption spectra associated with the fast evolving species (see, for example Fig. 4a, b) show only positive absorption bands

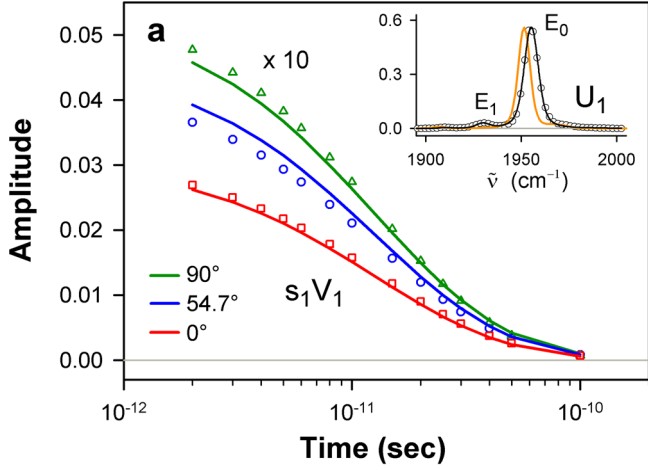

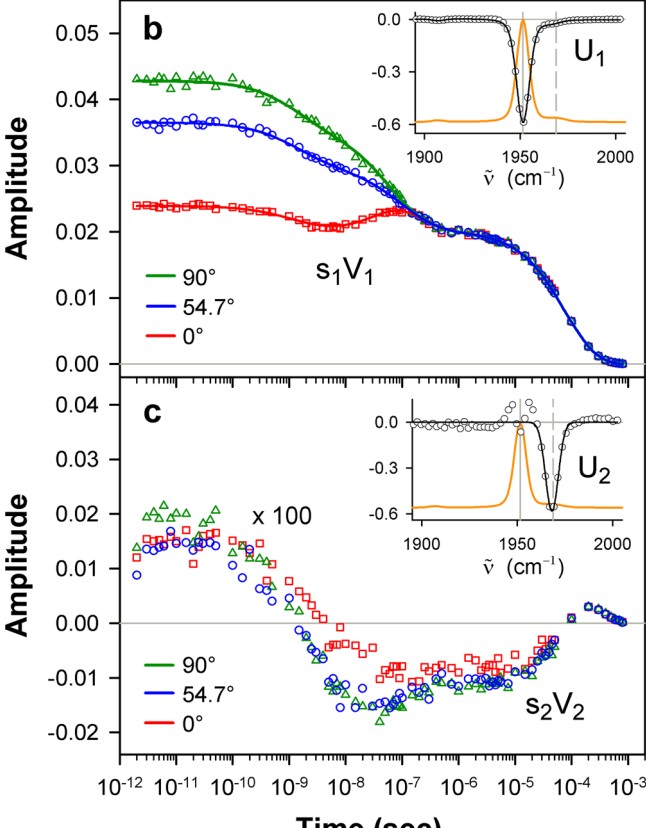

**Fig. 6 | Global analysis of the photoinduced mid-IR absorption changes of the isolated carbonmonoxy β chains in the region of ground-state bleach at the parallel, perpendicular, and magic angle polarization settings. a** Global SVD of the spectra associated with the intermediate species evolving in the first 100 picoseconds after the photoexcitation. **b, c** Global SVD of the spectra associated with the intermediate species evolving over a timescale ranging from nanoseconds to hundreds of microseconds. Global SVD in (**a**), as in (**b**), (**c**), was performed on the global data matrix composed of data obtained at three polarization settings. Description of (**a**) and (**b**) is the same as for Fig. 5. In (**a**), as in (**b**), the global fitting of the time-dependent amplitudes $s_1V_1$ at three polarization settings was performed as described in Supplementary Note 2. In (**c**), the time-dependent amplitude ($V_2$) of the second basis spectrum, multiplied by the corresponding singular value ($s_2$) is plotted at three polarization settings. The time course of $s_2V_2$ at the magic angle polarization setting is determined by (*i*) an overall decrease in amplitude due to the CO rebinding to the deoxygenated chains and (*ii*) spectral changes caused by the interconversion between the $A_0$ and $A_1$ conformational substates. In the inset to (**c**), the second basis spectrum ($U_2$) and its fit with a Gaussian function are shown in black as circles and the solid line, respectively. $U_2$ exhibits a band at 1968 cm$^{-1}$ near the stretching band of CO bound to the isolated Hb chains in the $A_0$ substate[45]. In all insets, the ground-state FTIR absorption spectrum normalized to an arbitrary value is shown as the orange solid line for comparison. In the insets of (**b**) and (**c**), the ground-state FTIR absorption spectrum is additionally offset for clarity. The positions of the vibrational bands of CO in the $A_0$ and $A_1$ substate[45] are shown as the vertical dashed and solid lines, respectively. Note the difference in the range for the horizontal axis in (**a**) vs. **b**, **c**. Source data are provided as a Source Data file.

and have no any sign of the ground state bleach recovery. The considered transient IR absorption spectra are described at the three polarization settings by only one basis spectrum, $U_1$ (Figs. 5a and 6a, insets). In $U_1$, two positive spectral peaks, designated hereafter as $E_0$ and $E_1$, are observed. The $E_0$ band is located ~3 cm$^{-1}$ higher in frequency than the main ground-state CO stretching band (~1950 cm$^{-1}$) for the corresponding carbonmonoxy-chain in the $A_1$ conformational substate (Supplementary Tables 3 and 6). Similarly, the $E_1$ band is located higher in frequency, at ~1930 cm$^{-1}$ (Supplementary Tables 6), than the 1→2 vibrational band for CO bound to the heme in the ground electronic state, as expected, at ~1925 cm$^{-1}$. This expected value was obtained from the 0→1 transition frequency using the anharmonic shift of 25 cm$^{-1}$ determined for the tetrameric HbCO in the ground electronic state[47]. As for the intensity, the relative integrated areas of the $E_0$ and $E_1$ bands are ~95% and ~5%, respectively. The spectral positions of these two bands being separated by 22–25 cm$^{-1}$ (Supplementary Table 6) are

in agreement with the anharmonic shift mentioned above for HbCO in the ground electronic state[47]. The average angle $\Theta$ between the CO ligand and the heme plane normal, determined here for the $E_0$ and $E_1$ states (Supplementary Table 5), is 20 ± 4°, being close to that for the heme-bound CO in the isolated carbonmonoxy chains in the $A_1$ conformational substate (previous Section).

To rule out the possibility that some photolyzed CO molecules can be responsible for the observed $E_0$ and $E_1$ bands, the orientation and spectral properties of the CO molecule in the $E_0$ and $E_1$ states should be compared with those of the photolyzed CO trapped in the primary docking site. It should be stressed that the orientation of the CO ligand in the $E_0$ and $E_1$ states differs significantly from that of the trapped CO molecule ($\Theta = 69° ± 6°$). Their vibrational spectra also differ significantly. The $E_0$ and $E_1$ bands are located ~180–205 cm$^{-1}$ lower in frequency than the $B_1$ band for the trapped CO molecule in the ground vibrational state (Fig. 3a, b). As will be shown below, it is virtually impossible that the $E_0$ and $E_1$ bands are due to transitions from higher excited vibrational levels of the trapped CO. For that to be the case, the trapped CO molecule would have to be in the seventh excited vibrational state ($v = 7$). This estimate was made taking into account the 0→1 transition frequency and the anharmonic shift of 27.5 cm$^{-1}$ for the photolyzed CO trapped in the primary docking site[63]. To excite the trapped CO to the seventh ($v = 7$) excited vibrational level, an energy of about 14,400 cm$^{-1}$ should be deposited into the vibrational mode of this molecule. In the present experiment, the $^1\pi\pi^*$(Q) state of the carbonmonoxy chains was directly excited at 543 nm (18,420 cm$^{-1}$). The bond energy required to break the bond between Fe and CO, which can be approximated by the measured enthalpy change for binding to the gaseous ligand[39,68], is ~23.1 kcal/mol (~8080 cm$^{-1}$) (averaged over α and β subunits)[69]. Therefore, at our photoexcitation conditions, ~40 % of the photon energy would be spent just on breaking the bond between the heme iron and CO. Hence, the remaining energy (~10,340 cm$^{-1}$) is not sufficient for the excitation of the trapped CO to the seventh ($v = 7$) excited vibrational level. Based on this conclusion and on the above-mentioned difference in the CO orientation, it is very unlikely that the $E_0$ and $E_1$ bands would correspond to highly excited vibrational states of the photolyzed CO trapped in the primary docking site. Moreover, it is unlikely that the $E_0$ and $E_1$ bands correspond to the lowest vibrational states of the photolyzed

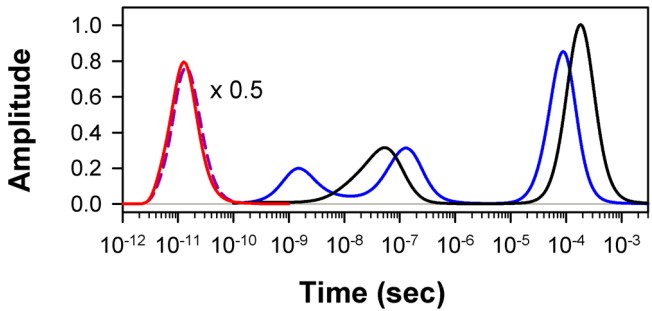

**Fig. 7 | Lifetime distributions.** The lifetime distributions describing the dynamics of the excited states deactivation in the isolated carbonmonoxy α and β chains are shown as the solid red line and the dashed violet line, respectively. The lifetime distributions describing the dynamics of the CO rebinding to the isolated α and β chains in the $A_1$ conformational substate are shown as the black and blue lines, respectively. Source data are provided as a Source Data file.

CO trapped in any secondary docking sites within the protein matrix, since, in this case, a very substantial downshift in the vibrational frequency is required to bring it to the similar value as for the heme-bound CO and nothing like that has been ever reported before. Therefore, we rule out that the positive transient $E_0$ and $E_1$ bands can originate from the photodissociated CO molecule and assign the $E_0$ and $E_1$ bands to the vibrational transitions of the CO ligand bound to the photoexcited heme.

In Figs. 5a and 6a, the first amplitude vector $s_1V_1$ for the magic angle polarization setting represents the kinetics of disappearance of the $E_0$ and $E_1$ bands. For each Hb chain, the MEM analysis of this $s_1V_1$ vector derived one lifetime distribution (Fig. 7, solid red and dashed violet line) with only one peak at -15 ps (Supplementary Table 5, τ). Both the $E_0$ and $E_1$ bands disappear simultaneously with the time constant of -15 ps. It should be stressed that, in the transient mid-IR absorption spectra associated with the intermediate species evolving in the first 100 picoseconds after the photoexcitation (the fast evolving species) (Fig. 4a, b), there is no any sign of a ground-state bleach recovery with such time constant. Such behavior is typical for the case when there is no return to the ground state after the photoexcitation within the considered time frame.

The present analysis of the experimental data reveals, along with the instantaneous depletion of the ground state of the CO-bound protein, the formation of an excited state of this liganded protein as well as the subsequent disappearance of the excited state on the picosecond timescale after the photoexcitation. Both observed absorption features $E_0$ and $E_1$ (Figs. 5a and 6a, insets), which disappear simultaneously with the same time constant (-15 ps), can be attributed to the vibrational bands of CO coordinated to the heme in this excited state. Obviously, the time constant of -15 ps characterizes the disappearance of that excited state. The recovery of the ground state of the liganded protein is solely determined by the geminate (nanosecond) and bimolecular (microsecond) CO rebinding processes, which are up to three orders of magnitude slower than the disappearance of the excited state of the liganded protein (-15 ps time constant). The observed discrepancy between the time constants for the disappearance of the excited state (Fig. 7, solid red and dashed violet line) and those for the recovery of the ground state of the liganded protein (Fig. 7, black and blue line) unambiguously indicates the absence of a direct transition from this excited state to the ground state of the carbonmonoxy-heme proteins. The only possible channel for the disappearance of the excited state of the liganded protein is the CO photodissociation with the -15 ps time constant. This slow (-15 ps) CO photodissociation is in an agreement with the observed increase in the total integrated area of all the transient bands corresponding to the photolyzed CO (Fig. 3c, d). It should be emphasized that the vibrational

relaxation of the photolyzed CO trapped inside the protein matrix is much slower (hundreds of picoseconds)[64] than the discussed slow (-15 ps) CO photodissociation process and therefore does not interfere with our interpretation of the increase in the total integrated area. The above-mentioned fact that the ground state bleach recovery is solely determined by the CO rebinding means that every photon absorbed by the heme leads to the Fe−CO bond breaking, consistently with the quantum yield value close to unity for the CO photodissociation[11].

## Discussion

The CO photodissociation with the -15 ps time constant, observed in the present work, occurs in addition to the main prompt sub-50-fs CO photodissociation process known in the literature[30]. The breaking of the bond between the heme iron and CO is not a single-step process, but rather at least a two-step process extended in time from tens of femtoseconds to tens of picoseconds. The observed ground-state bleaching is due to the depletion of the ground state resulting in population of excited states, followed by both the fast (<50 fs) and slow (-15 ps) CO dissociation. The contribution of both fast and slow photodissociation events to the overall photodissociation process can be estimated by taking into account that (i) the area of the bands in the ground-state bleach spectra is proportional to the concentration of both rapidly and slowly dissociating species and that (ii) the area of the $E_0$ and $E_1$ vibrational bands is proportional to the concentration of only slowly dissociating species. Assuming similar extinction coefficients for these vibrational bands and those in the ground-state bleach spectra, the contribution of the slow photodissociation to the overall photodissociation process can be estimated from the magic angle data as the ratio of the areas of the corresponding bands at the initial time moment. This ratio is equal to 0.11 for both Hb chains. Based on this value, the contributions of the fast and slow photodissociation events to the overall photodissociation process differ significantly and are 89% and 11%, respectively.

Direct observation of the two-step photodissociation of CO in the present work is the experimental proof for the existence of several dissociation channels involved in the CO photodissociation. In order to explain our experimental data, we propose a model for the Fe−CO photolysis that includes successive population of two dissociative states after the photoexcitation (Fig. 8). Here, $HbCO(^1A_1)$ denotes the carbonmonoxy chains with the heme in the ground electronic state, while Hb denotes the deoxy chains with the heme in the ground as well as in electronic and/or vibrationally excited states formed after the CO photodissociation. These excited states of the deoxy-species have been intensively discussed in the literature[12,30–32]. Within the framework of our model, it is assumed that after the photoexcitation into the Q band of the $^1π–π^*$ nature (Fig. 8, $HbCO(^1Q)$), an excited electronic state of the liganded protein, $HbCO^*_I$, is initially populated. Subsequently, either the fast step of photodissociation proceeds with the time constant $τ_I$ or a transition to another low-lying excited state of the liganded protein, $HbCO^*_{II}$, occurs with the time constant $τ_{tr}$. The probabilities of these two processes are equal to the contributions of the fast and slow photodissociation to the overall photodissociation process, being 89% and 11%, respectively. The ratio of these probabilities indicates that the CO dissociation in the excited state $HbCO^*_I$ must be 8 times faster than the transition from $HbCO^*_I$ to $HbCO^*_{II}$. Taking the value for $τ_I$ as -50 fs[30], this gives the time constant $τ_{tr}$ to be -400 fs. In the low-lying excited state $HbCO^*_{II}$, the slow step of photodissociation, discovered in the present study, occurs in $τ_{II} ≈ 15$ ps, three hundred times slower than that in the initially populated $HbCO^*_I$ state ($τ_I$-50 fs).

A similar -300 fs value (as $τ_{tr}$-400 fs obtained here) was reported earlier for a relaxation process in the heme proteins after the photoexcitation by the visible light[12,30] and eventually assigned to the transition between different excited electronic states of the heme[12]. The time constant of -15 ps for the disappearance of the $HbCO^*_{II}$ state is

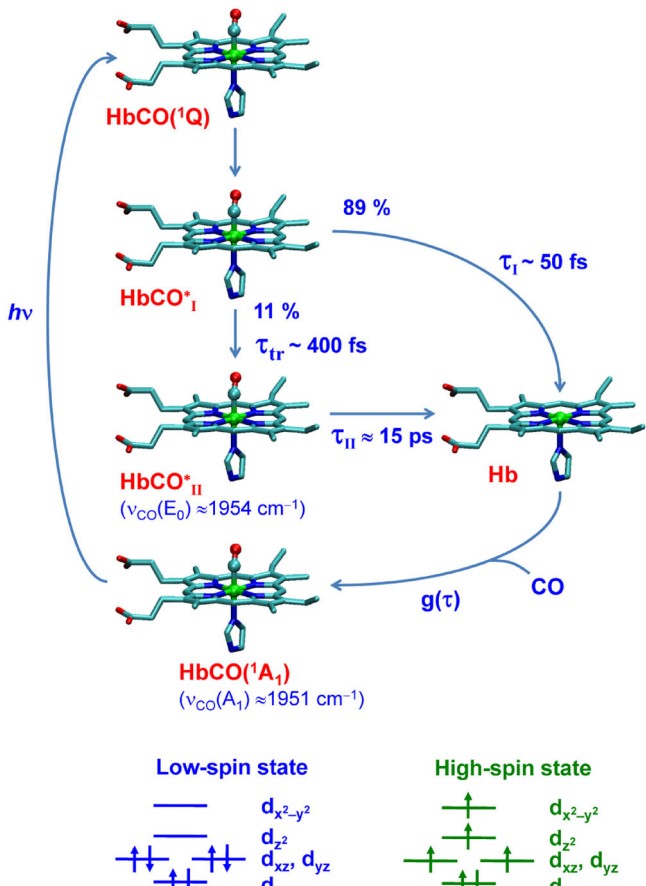

**Fig. 8 | Schematic representation of the two-channel photodissociation of CO from ferrous HbCO.** The excited-state species HbCO*$_I$ and HbCO*$_{II}$ are assumed to be formed sequentially from the ground-state species HbCO($^1$A$_1$) after the photo-excitation to the $^1$ππ*(Q) state, HbCO($^1$Q), of the CO-bound protein. τ$_{tr}$ is the time constant for the transition from HbCO*$_I$ to HbCO*$_{II}$. τ$_I$ and τ$_{II}$ are the time constants for the formation of the deoxy-like species Hb from HbCO*$_I$ and HbCO*$_{II}$, respectively. g(τ) is the lifetime distribution describing both the geminate and bimolecular ligand rebinding to the heme protein. To simplify the diagram and focus only on the CO photodissociation, protein conformational transitions are not shown. The diagram is labeled for the case of Hb, but it is generally applicable for each Hb subunit in a chosen conformational substate. For the α and β chains in the A$_1$ conformational substate, the lifetime distributions g(τ) are shown in Fig. 7 (black and blue line, respectively). The A$_1$ conformational substate is characterized by the infrared absorption band of the bound CO, ν$_{CO}$(A$_1$) ≈ 1951 cm$^{-1}$; the E$_0$ state, ν$_{CO}$(E$_0$) ≈ 1954 cm$^{-1}$. The ground state of HbCO and Hb is a singlet ($^1$A$_1$) and quintet ($^5$B$_2$), respectively. The corresponding ground state electronic configurations for the ferrous iron ion are shown at the bottom of the figure (low- and high-spin state, respectively).

close to that of 18 ± 2 ps determined by two-color IR pump – IR probe spectroscopy for the vibrational relaxation of the CO stretch vibration in HbCO in the ground electronic state[70]. The vibrational relaxation of the heme upon photodissociation of the CO ligand is also on the order of 10–20 ps[33]. Despite the comparable timescale of these cooling processes, the complete lack of the matching ~15 ps component in the recovery of the ground state bleach rules out the vibrational cooling as the origin of the ~15 ps process observed in our study.

In the heme proteins, ultrafast energy redistribution over different vibrational modes has been thoroughly studied[31]. It was established that, after absorption of the visible light, energy is transferred promptly to a highly excited vibronic manifold by Frank-Condon coupling. From this manifold, the vibrational energy is transferred with different timescales into low- and high-frequency modes, prior to slow

dissipation through the protein on the ~10−20 ps timescale. In the ground electronic state, vibrationally excited levels of high-frequency modes (v > 1000 cm$^{-1}$) are populated in a few hundred femtoseconds, and relax in several picoseconds after the photoexcitation[31]. In contrast, vibrationally excited levels of low-frequency modes are populated quasi-instantaneously and are thermally equilibrated within ~300 fs[31]. Taking into account that molecules such as hemes contain about one hundred vibrational modes, it is unlikely that, at 2 ps after the photoexcitation (the earliest time delay considered in our data analysis), a large amount of excess energy (up to ~40% of the absorbed photon energy, ~8080 cm$^{-1}$) required to break the Fe−CO bond in the ground electronic state is still accumulated in a single vibrational mode such as the dissociative Fe−CO stretching mode at ~500 cm$^{-1}$ [12]. Therefore, it is unlikely that the observed slow (~15 ps) CO photodissociation can be caused by vibrationally excited states of the heme −CO complex in the ground electronic state. It is unlikely that both the E$_0$ and E$_1$ states are the vibrationally excited states of the ground electronic state. The fact that the E$_0$ and E$_1$ bands (Figs. 5a and 6a, insets) appear several wavenumbers higher in frequency than the heme-bound CO stretch bands (the 0→1 and 1→2 vibrational transitions, respectively) for the carbonmonoxy heme proteins in the ground electronic state rules out the assignment of the E$_0$ and E$_1$ bands to the vibrationally excited CO in the heme-CO complex in the ground electronic state. Shelby, Harris, and Cornelius predicted in their model[71] that the vibrational frequencies for polyatomic molecules may shift to higher wavenumbers owing to the anharmonic coupling with other vibrational modes. However, such anharmonic coupling would inevitably result in the corresponding bleach of the vibrational ground state (as typically observed in 2DIR experiments), which has to recover on the same timescale as the decay of the E$_0$ and E$_1$ bands, which we do not observe in the present case. The above-mentioned conclusion concerning the impossibility of the CO dissociation in the vibrationally excited states of the ground electronic state at our photoexcitation conditions is indirectly confirmed by experiments with direct mid-IR excitation of the CO stretch vibration in the ground electronic state of HbCO, when up to six excited CO vibrational levels were populated[47]. In those experiments, it was tested if the Fe−CO bond breaking could be caused by an energy flow from the initially excited high-frequency CO stretch mode into the low-frequency Fe−CO stretch mode (if the coupling between the CO and the FeC bonds was sufficiently strong). However, no marked CO photodissociation was observed indicating that fast dissipation of the vibrational energy from the Fe−CO mode prevents the population of highly excited vibrational levels of that mode. Based on the discussion above, we concluded that the slow (~15 ps) CO photodissociation, discovered in the present study, proceeds in an electronically excited state of the heme−CO complex. Taking into account the relative integrated areas of the E$_0$ and E$_1$ bands, we assign the E$_0$ and E$_1$ bands, respectively, to the 0→1 and 1→2 vibrational transitions of the CO ligand bound to the heme in this long-lived dissociative excited state. In Figs. 5a and 6a (insets), the observed shift of the 1→2 vibrational transition (E$_1$ band) to lower wavenumbers relative to the 0→1 transition (E$_0$ band) is typical due to the anharmonicity of the vibrational potential[72].

The mechanism of the heme−CO photolysis has been elucidated recently by means of high-dimensional quantum dynamics calculations[35]. It has been shown that after photoexcitation to the $^1$Q state, an excited singlet metal-to-ligand-charge-transfer (MLCT) state is populated[35]. Strong Jahn-Teller distortions in this state afford an efficient energy transfer from the porphyrin plane (x, y axes) to the Fe −CO axis (z axis), activating dissociative stretching vibrations, which can lead to CO dissociation[35]. Then, in the heme−CO complex, spin crossover occurs. After ~75 fs, the $^3$MLCT manifold is clearly dominant, although a residual $^3$MC participation is also observed. From the triplet manifold, the quintet excited states, $^5$MLCT and $^5$MC, are populated, becoming dominant at around 400 fs. It should be noted that the sub-

picosecond timescale of photoinduced spin crossover in 3d[6] ferrous compounds has also been established experimentally by femtosecond X-ray emission and scattering spectroscopy[36,73,74]. The value at around 400 fs, obtained by quantum dynamics calculations[35] for the population of the quintet states, [5]MLCT and [5]MC, matches perfectly the time constant for the transition from HbCO*$_I$ to HbCO*$_{II}$, $\tau_{tr}$~400 fs, estimated in the present work. Therefore, we tend to assign the HbCO*$_{II}$ state introduced in our model to the [5]MLCT/[5]MC excited states. As has been noted in ref. 35, the $\sigma-\pi$ bond between Fe and CO is weakened for the MLCT states. The [5]MC excited states, being the lowest excited electronic states in the heme − CO complex[35], involve transitions from the occupied $d_{xy}$, $d_{xz}$ and $d_{yz}$ orbitals to the unoccupied $d_z^2$ and $d_{x^2-y^2}$ orbitals. Promoting an electron into the antibonding $d_z^2$ orbital would increase the Fe−CO distance and greatly reduce the synergic $\pi$-back-bonding, on which the stability of the Fe−CO bond depends. This strengthens the assumption that the [5]MLCT/[5]MC excited states could be those dissociative ones in which the CO photodissociation occurs with the ~15 ps time constant. Both the prompt sub-50-fs and slow ~15 ps dissociation steps together result in the photoinduced Fe−CO bond breaking efficiency of nearly unity.

In this work, we report the experimental proof for the two-step photodissociation of CO from the isolated Hb chains. The breaking of the chemical bond between the ferrous heme iron and CO is found to involve both the prompt sub-50-fs process previously known in the literature and the additional slower ~15 ps process discovered in this study. The contribution of the fast and slow photodissociation events to the overall process was estimated to be 89% and 11%, respectively. The model of Fe−CO photolysis that includes successive population of the two dissociative states, HbCO*$_I$ to HbCO*$_{II}$, is introduced and discussed. This work illustrates the power of the time-resolved multiple-probe mid-IR spectroscopy for unraveling the exquisite details of the ligand release and uptake in heme proteins. The discovery of the long-lived dissociative state significantly changes current understanding of the mechanism of Fe−CO photolysis in heme proteins and provides essential insight into the small molecules transport in physiological systems.

## Methods

### Sample preparation
The isolated Hb chains in the carbonmonoxy form were prepared according to the procedure described in detail in Supplementary Note 3. All the spectroscopy experiments were performed in 50 mM deuterated Tris buffer, pD 8.2, at 19 °C. D$_2$O was used to avoid strong water absorption in the spectral region of interest. Concentrations of the isolated carbonmonoxy $\alpha$ and $\beta$ chains were 3.0 and 4.0 mM in heme, respectively. The protein samples were checked before and after the experiment by UV-Vis and FT-IR spectroscopy.

### Time-resolved mid-IR spectroscopy
Picosecond to millisecond time-resolved transient absorption spectra in the mid-IR region were measured on the ULTRA[60] apparatus in the Time-Resolved Multiple Probe Spectroscopy mode[61] at the Central Laser Facility (STFC Rutherford Appleton Laboratories, Harwell, UK). The time-resolved multiple-probe spectrometer comprised two Ti:sapphire amplifiers with different repetition rates (1 and 10 kHz, respectively). The 1 kHz Ti:sapphire amplifier (Spitfire XP, Spectra-Physics, USA) was used to pump the OPA (TOPAS, Light Conversion, Lithuania) to provide 1 kHz, 100 fs, 543 nm pump pulses. These pulses were attenuated to deliver 1 μJ at the sample. The pump laser fluence (approximately 5.7 mJ·cm$^{-2}$) was within the linear photoexcitation regime[75]. The 10 kHz Ti:sapphire amplifier (Thales Laser), producing 0.8 mJ output with 40 fs pulse duration, at 800 nm, was used to pump another OPA (TOPAS, Light Conversion, Lithuania) to generate mid-IR pulses. The mid-IR beam was split to provide the probe beam passing through the sample and the reference beam

bypassing the sample. The probe and reference beams were dispersed with home-built spectrographs and detected using HgCdTe arrays (IR Associates). High-speed data acquisition systems (Quantum Detectors) allowed 10 kHz acquisition and processing of the probe and reference pulses to generate a pump-on/pump-off IR absorption difference signal. The mid-IR transient absorption spectra were calibrated using the characteristic absorption bands of polystyrene and 1,4-dioxane.

Both the 1 and 10 kHz amplifiers were optically synchronized by sharing the same seed from the Ti:sapphire oscillator (Synergy, Femtolaser, Germany) operating at 68 MHz and delivering 20 fs pulses. The arrival time of the seed beam to the 1 kHz amplifier was adjusted with a tunable optical delay line to achieve the time delay between the pump and probe pulses over the range of 100 fs–14.7 ns. To go beyond 14.7 ns up to 100 μs, subsequent oscillator seed pulses were selected, accompanied by the appropriate setting of the optical delay line. The 10 kHz probe pulses, continuously arriving to the sample every 100 μs, cover the time range up to 1 ms between the subsequent excitation pulses.

The pump and probe beams were focused at the sample into ~150 and 80 μm spots (FWHM), respectively. The relative polarization $P$ of the pump and probe beams was switched during the experiment between the parallel (0°), perpendicular (90°), and "magic" angle (54.7°) configurations. Samples were raster scanned in the $x$ and $y$ directions to minimize photodamage and re-excitation effects. The sample solutions were contained in demountable IR cells (DLC-S13, Harrick Scientific Products) with CaF$_2$ windows (13 mm diameter, 2 mm thick) and 100 μm optical path length. During the measurements, the time delays were scanned in a random order to avoid systematic effects in the data due to possible sample degradation. At each time delay, the present spectra were collected for 3 seconds accumulation time. The data acquisition was repeated 5 times to average the data. The time-resolved spectrometer[60] used here allows to detect transient absorption spectra in the region of interest (1880–2160 cm$^{-1}$) in the time range from 2 ps up to 800 μs after the laser photoexcitation with a sensitivity of 10$^{-5}$ absorbance units. A baseline was subtracted from the spectra. For each protein, differences in the amplitudes of the spectra obtained at the three polarizations due to variations in the level of photolysis were eliminated by matching the amplitudes at times long enough compared to the rotational correlation time of the molecules. The degree of photolysis in all the experiments did not exceed 20%.

### Reporting summary
Further information on research design is available in the Nature Portfolio Reporting Summary linked to this article.

## Data availability
Source data underlying the figures in this paper are provided as a Source Data file. The time-resolved transient IR absorption spectra generated in this study are openly available from eData the STFC Research Data repository at [https://doi.org/10.5286/edata/947]. Source data are provided with this paper.

## Code availability
In the present work, the singular value decomposition (SVD) analysis was applied, the algorithm of which is built into many programs, including the IgorPro program, which was used in this case. For the maximum entropy method (MEM) analysis, the program MemExp (version 3.0) was used and cited in Supplementary Information (refs. 3 and [4]).

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

## Acknowledgements

The authors are greatly indebted to Dr G. M. Greetham, Dr P. M. Donaldson, Dr E. Gozzard, and B. Landowski for their help in setting up the experiment. Dr P. J. Steinbach is kindly acknowledged for the use of MemExp. S.V.L., I.V.S., and M.V.P. thank STFC for the beamtime on the CLF Ultra facility (App. 16130005). This work was supported by the Belarusian Republican Foundation for Fundamental Research (Grant No. F19MS-009) (S.V.L., M.V.P., J.A.W., and B.M.D.) and the National Academy of Sciences of Belarus through the Program "Photonics and Electronics for Innovation 2021–2025" (Project 1.8) (S.V.L., M.V.P., and B.M.D.).

## Author contributions

S.V.L., I.V.S., and M.V.P. designed the experiment. S.V.L., M.V.P., S.N.G., A.V.Y., and J.A.W. were involved in the preparation of the protein samples. S.V.L., I.V.S., and M.T. performed the transient absorption experiments. S.V.L. processed and analyzed the data, wrote the manuscript. S.V.L., I.V.S., M.V.P., J.A.W., M.T., and B.M.D. discussed the results and the data analysis and contributed to the data interpretation. All authors contributed to reviewing the manuscript.

## Competing interests

The authors declare no competing interests.
