## [Transparent Peer Review file · Nature Communications]

Direct observation of two-channel photodissociation of carbon monoxide from the hemoglobin subunits

Corresponding Author: Dr Igor Sazanovich

Version 0:

Reviewer comments:

Reviewer #1

(Remarks to the Author)

In this study, Lepeshkevich and coworkers use picosecond time-resolved mid-IR spectroscopy to provide evidence of a new rate of 15 ps, attributed to a photolysis channel of carbon monoxide from heme proteins. While this channel is minor (they attribute that 11% undergo such dissociation), it is important whether there are alternative dissociation channels besides the sub-picosecond dissociation.

The main observation band is the CO stretching around 1925-1950 cm^{-1} , which the authors attribute to the stretching of CO when bound to Fe. The observation of a decay of such a band makes them to attribute it to the new dissociation channel, which is further supported by a hypothetical band structure predicted by Eaton and coworkers in 1978 using absorption spectroscopy and crystal field theory.

At the current stage, the authors do not provide a sufficiently convincing explanation therefore, I would suggest a major revision of the current work, reconsidering the following points:

- 1) The reaction of CO dissociation is known to depend on the pump power, as recently shown by Schlichting et al., Nature, 2024. This study is of utmost importance here, as multiphoton absorption has been shown to accelerate the dissociation of Fe-CO. Using the tr-mid-IR technique, and if the interpretation of the 1950 cm^{-1} band is correct, they should not see any signal at high pump fluences.
- 2) The supposition that the shifts of the CO bands are like the ground-state, has been proven to be wrong. The photodissociation of CO is so fast, that the CO obtains a much larger vibrational energy. The decays that the authors observe could be simply different decay channels of photolyzed CO^* to the protein. There should be better proof (from theory or some model system) that the band at 1950 cm^{-1} corresponds to an excited state-bound Fe-CO.
- 3) There is a complete lack of literature review from time-resolved mid-IR. Several studies in the past have tried to dissociate Fe-CO by multiple IR photon absorption of 1950 cm^{-1} in the ground state. Even 6 photon absorption did not manage to dissociate in the ground state. In the excited triplet $3T_1$, the bond is weaker, so how many quanta of vibrational energy is needed to break the triplet Fe-CO bond? This should be estimated.
- 4) How is the energy transferred to CO modes? The absorption occurs in the heme xy-plane, and then is transferred to the z direction (His-Fe-CO bonds). There is no direct vibrational coupling between heme and CO, so the energy needs to flow via Fe-CO to the CO modes. If CO is active, it means that Fe-CO modes have a lot of energy and thus dissociation should occur before. Perhaps, since this process is very fast, again, there are simply different excited unbound CO^* channels that dissipate energy to the protein.
- 5) The calculations of Eaton and coworkers are not sufficient for the interpretation of an extra channel for dissociation. They were simply based on crystal field theory, and Kasha's rule, meaning that dissociation could potentially occur from the lowest singlet and the lowest triplet states. The main dissociation channel in sub-picosecond is clearly from singlet, but not $1T_1$ as described by Eaton et al. Recent theory showed that indeed, 10-20% of the population ends up in the lowest metal-centered, but it is not shown to be dissociative. A further mechanistic explanation using modern theoretical techniques of such a dissociation channel is currently lacking and needed for the correct interpretation of the observed data.

Reviewer #2

(Remarks to the Author)

The authors reported CO photodissociation from the long-lived excited state(s) of HbCO and discussed the mechanism of heme–CO photodissociation based on unique picosecond to millisecond time-resolved mid-IR data. They attributed the observed spectral changes to the two-step photodissociation of CO and claimed that this is the first experimental proof of the existence of several dissociative states of HbCO. The results of this study will likely be of great interest to the readership of this journal. The interpretation of the experimental data may need to be reconsidered. Therefore, I recommend that the authors make appropriate revisions to publish this manuscript.

The authors detected a 15 ps component in the IR absorption spectral change by careful measurements. The quality of the measured data is high. It is convincing that this 15 ps component is not an artifact. However, the conclusion that the breaking of the chemical bond between ferrous heme iron and CO includes a slower (~15 ps) process cannot be drawn from the experimental results reported in the manuscript by following reasons.

The authors ruled out the possibility that E0 is the vibrationally excited state of the ground electronic state, as the vibrational excited states are typically shifted to a lower frequency relative to the fundamental frequency. However, in the model by Shelby, Harris, and Cornelius, the transition frequencies of the vibrationally excited states may shift to higher wavenumbers owing to the anharmonic coupling with other vibrational modes.

R. M. Shelby, C. B. Harris, P. A. Cornelius; The origin of vibrational dephasing of polyatomic molecules in condensed phases. *J. Chem. Phys.* 1 January 1979; 70 (1): 34–41.

Therefore, it is possible that CO stretching vibrations exhibit higher wavenumber even in the vibrationally excited state of the electronic ground state.

The authors observed that a small increase (up to 20 %) of the total integrated area occurring with a time constant of ~20 ps (Supplementary Fig. 4c, d) and attributed it to the slow dissociation of CO. However, such an intensity increase can be observed for the vibrational relaxation of a dissociated CO molecule if the absorption cross-section of CO depends on the vibrational quantum number.

Therefore, both experimental results can be explained by the vibrational relaxation of CO

If the authors' interpretation is correct, an electronic absorption band originating from the heme of HbCO*(II) should appear after photoexcitation, which should decay at ~15 ps. Is such an electronic absorption band not observed?

Reviewer #3

(Remarks to the Author)

Direct observation of two-channel photodissociation of carbon monoxide from the hemoglobin subunits

Context : The present work is focused on the photodissociation of the diatomic CO (carbon monoxide) from the heme of isolated subunits of hemoglobin. The authors employed pump-probe infra-red absorption spectroscopy in the 1 ps to 1 ms time range to measure the evolving population of CO photodissociated from the heme. The IR probing frequency is that of heme- liganded CO (1950 cm⁻¹) and dissociated CO (2125 – 2135 cm⁻¹) and we may consider the phenomenon to be probed in the « ligand point of view ».

Results : In the ps time range the populations of liganded CO and dissociated CO both show an evolution with a 15-ps time constant which is assigned to a second dissociation channel (11%) in addition to the main fast one (≤50 fs, 89%), thus from two distinct electronic excited states of the CO-liganded heme. In the slower time range, the kinetics of geminate (ns-μs) and bimolecular (ms) rebinding of CO are well resolved. They appeared different for the α and β isolated subunits of Hb.

Rationale : The existence of the second slower photodissociation channel (15 ps) is based on the evolution of the CO populations. From its existence and previous studies, a model for electronic pathways of heme excited states leading to ligand dissociation is drawn.

The photodissociation of diatomics from the heme is indeed not fully explained and the existence of two electronic pathways is an important information. The data presented here are of good quality and rigorously analyzed. They support the conclusion and provide a proof of the existence of two photodissociation channels. I recommend publication of the present manuscript.

I do not have concerns, only a few comments and questions :

The authors wrote at the end of introduction (page 5 and also in page 12) that Eaton et al. (Ref. 31) have « first predicted the existence of several dissociative states of HbCO. » However, whereas Eaton et al. hypothesized which excited state could

lead to dissociation (several are possible), they did not mean that several channels are simultaneously used. Also, Eaton et al. estimated an « upper limit for the dissociative time constant of 16 ps » for HbCO. We must note that this value was only based on the calculation of quantum yield (not on kinetics) which was very approximate since they used a high energy 8-ps excitation pulse at 353 nm (28328 cm⁻¹) for dissociating HbCO. Thus the value of 16 ps given by Eaton et al. should not be considered as a prediction of that measured in the present work.

I suggest to the authors to incorporate Supplementary Figure 4 in the main manuscript (at least panels a and b) because these data strengthen those given in Fig. 2 and are necessary. Indeed, the increase of the induced absorption is a strong proof and cannot be dissociated from the evolution of the bleaching.

In Fig. 5a the fitted kinetics do not strictly follow the experimental points. Is this due to the fact that the authors imposed exactly the same time constant for fitting the three polarized kinetics ?

Contrarily, the fitted kinetics in the Supplementary Figures go through the experimental points.

Can the authors amend the figure or comment ?

The authors calculated a 400-ps time constant for the transition 1T1 to 3T1 states. It should be detectable by transient electronic absorption. For example in Ref. 26 a 300-ps component was reported, although assigned differently as here. Can the authors shortly comment on the assignment in Ref. 26 ?

The spectral SVD component U2 represents the small shift of the absorption band E0 in the slow dissociation path with respect to the fast one. It is also discernable on Fig. 2 b and e. I agree with the authors' justification to discard an electronic ground state origin. How to insert this information within the frame of the model of Fig. 7 ?

Although the authors provide a Ref. (61) to describe the difference in the geminate CO rebinding rate between α and β chains, they did not give explanation. Since the data are new, it is worth to add at least a sentence for providing a structural explanation.

Version 1:

Reviewer comments:

Reviewer #1

(Remarks to the Author)

All my previous comments have been satisfactorily addressed. I recommend the manuscript for publication in the current form without any further revisions.

Reviewer #2

(Remarks to the Author)

The authors have made appropriate revisions to the manuscript. The manuscript is now considered acceptable for publication.

Reviewer #3

(Remarks to the Author)

The authors have fully and satisfactorily addressed all the concerns and questions raised in the previous review. I thus recommend publication without further change.

Michel Negrerie, PhD.

Dr. Igor V. Sazanovich
Central Laser Facility
STFC - Rutherford Appleton Laboratory
Harwell Campus, Chilton
OX11 0QX

The Reviewers
Nature Communications

31st of March 2025

Re: Response to the reviewers (NCOMMS-24-67838):

Dear Reviewers,

We wish to thank you for the very valuable comments and suggestions which undoubtedly improve the quality of our paper.

Below we summarize the changes we have made to our paper to address all those comments and suggestions. We hope the reviewers will be satisfied with our response, as we believe we have composed a strong and convincing story of the two-channel ligand photodissociation of carbon monoxide from heme proteins.

Before we go to the detailed point-by-point response to the comments and suggestions raised by the reviewers, we would like to outline very briefly only the key experimental evidence and how we interpret it to put our response into perspective:

1. We detect the instant bleaching of the ground state vibrational band of the heme-bound CO.
2. We observe new transient vibrational bands E_0 and E_1 which decay with 15 ps lifetime. The E_0 is shifted by 3 cm^{-1} **higher in frequency** with respect to the ground state vibrational band of heme-bound CO. The frequency of E_0 band is much lower than the known frequency of free CO trapped within the protein (see below) however is very close to the frequency of heme-bound CO in the ground state. The E_1 transient band is down-shifted by $22\text{--}25\text{ cm}^{-1}$ from E_0 . 25 cm^{-1} is a well-known anharmonic shift for the heme-bound CO vibration which points to E_1 being the $1\rightarrow 2$ transition of the same species as those of E_0 .
3. We observe no evidence of the bleach recovery for the ground state vibrational band of heme-bound CO on the picosecond time scale until the nanosecond ligand rebinding begins. Therefore the observed 15 ps process of E_0 and E_1 bands decay **is not ending in the ground state** of heme-bound CO.
4. We observe the vibrational signature of the photodissociated CO trapped within the protein matrix, located about 180 cm^{-1} higher in frequency than the ground-state vibration of the heme-bound CO. The vibrational signal of the photodissociated CO shows the presence of the vibrationally excited (to $v = 1$) unbound CO and this vibrationally excited unbound CO signal persists on the time scale much longer than 20 ps in accord with the literature data for the vibrational relaxation rate of the photodissociated CO trapped within the protein matrix. **More importantly**, we observe the overall grow-in of the signal of the photodissociated unbound CO

with the time constant of ~ 20 ps (much shorter than the vibrational cooling of the photodissociated unbound CO). This ~ 20 ps grow-in of the photodissociated unbound CO matches rather well (within experimental error) with the 15 ps decay of E_0 and E_1 transient CO bands.

5. From our anisotropy measurements for the new transient bands E_0 and E_1 , the angle between the CO and the heme plane normal is $20 \pm 4^\circ$, which is similar to such angle for the heme-bound CO in the ground state ($17 \pm 1^\circ$) however is very different to such angle for the photodissociated unbound CO ($69 \pm 6^\circ$) obtained in this work. This indicates that E_0 and E_1 highly likely correspond to heme-bound CO however in a state different to the ground state. At the same time E_0 and E_1 **are unlikely** to be the vibrations of the heme-bound CO in the vibrationally hot ground electronic state of the heme, as we **observe no bleach recovery** of the ground state vibrational band of the heme-bound CO until the nanosecond ligand rebinding begins.

As we have substantially revised our paper, we explicitly specify in the response below which parts of the paper text have been modified and how. Also we submit the two sets of files for the manuscript and Supplementary Information. In one set of files we highlight with coloured font all the changes made in the course of the revision with respect to the original version, and the other set of files has no colour-coding.

And now we provide the detailed point-by-point response to the comments and questions raised by the reviewers. We use the font colour and style to highlight the reviewer's text and our responses. The original reviewer's text is provided in *blue italic font*, whereas our responses are provided in black normal font.

Reviewer #1 (Remarks to the Author):

In this study, Lepeshkevich and coworkers use picosecond time-resolved mid-IR spectroscopy to provide evidence of a new rate of 15 ps, attributed to a photolysis channel of carbon monoxide from heme proteins. While this channel is minor (they attribute that 11% undergo such dissociation), it is important whether there are alternative dissociation channels besides the sub-picosecond dissociation.

The main observation band is the CO stretching around $1925\text{-}1950\text{ cm}^{-1}$, which the authors attribute to the stretching of CO when bound to Fe. The observation of a decay of such a band makes them to attribute it to the new dissociation channel, which is further supported by a hypothetical band structure predicted by Eaton and coworkers in 1978 using absorption spectroscopy and crystal field theory.

At the current stage, the authors do not provide a sufficiently convincing explanation therefore, I would suggest a major revision of the current work, reconsidering the following points:

1) The reaction of CO dissociation is known to depend on the pump power, as recently shown by Schlichting et al., Nature, 2024. This study is of utmost importance here, as multiphoton absorption has been shown to accelerate the dissociation of Fe-CO. Using the tr-mid-IR technique, and if the interpretation of the 1950 cm^{-1} band is correct, they should not see any signal at high pump fluences.

Response: We thank the reviewer for bringing up the pump power dependence of the CO dissociation. Indeed in the paper by Schlichting et al., Nature, 2024, the related heme protein myoglobin was studied in the similar range of concentrations as used by us, and Schlichting et al. confirmed the laser pump fluence for the single-photon excitation regime to be below $10\text{ mJ}\cdot\text{cm}^{-2}$. In our study we utilised exactly that excitation regime by keeping the pump fluence at $\sim 5.7\text{ mJ}\cdot\text{cm}^{-2}$ and the pulse length used was approximately twice as long as compared to the paper by Schlichting et al., Nature, 2024. Therefore for our experimental conditions, the possibility of multiple heme excitations is negligible.

To make it explicit, we have added the following statement in the “Methods” section (“Time-resolved mid-IR spectroscopy”) on Page 19:

”The pump laser fluence (approximately $5.7 \text{ mJ}\cdot\text{cm}^{-2}$) was within the linear photoexcitation regime.⁷⁵”, and also added the corresponding reference:

75. Barends, T. R. M. et al. Influence of pump laser fluence on ultrafast myoglobin structural dynamics. *Nature* **626**, 905–911 (2024).

As the reviewer correctly pointed out, at high pump fluence, the heme would experience multiphoton absorption, which is expected to accelerate the dissociation of the ligand as multiphoton excitation changes the nature of the dissociative excited states. This is a very intriguing idea, which we aim to explore in our future work.

2) The supposition that the shifts of the CO bands are like the ground-state, has been proven to be wrong. The photodissociation of CO is so fast, that the CO obtains a much larger vibrational energy. The decays that the authors observe could be simply different decay channels of photolyzed CO to the protein. There should be better proof (from theory or some model system) that the band at 1950 cm^{-1} corresponds to an excited state-bound Fe-CO.*

Response: We thank the reviewer for pointing out how we can strengthen our discussion of the origins of the detected E_0 and E_1 transient bands. We have substantially revised the final paragraph in the “Results” section, on Pages 11–13. We have re-named this paragraph which is now titled ”Long-lived dissociative excited state”. On Page 11–12 we have strengthened our reasoning to prove that the E_0 and E_1 bands do not originate from the photodissociated CO molecule and emphasized our assignment of the E_0 and E_1 bands to the vibrational transitions of the CO bound to the photoexcited heme. The text we have added now reads as follow:

“To rule out the possibility that some photolyzed CO molecules can be responsible for the observed E_0 and E_1 bands, the orientation and spectral properties of the CO molecule in the E_0 and E_1 states should be compared with those of the photolyzed CO trapped in the primary docking site. It should be stressed that the orientation of the CO ligand in the E_0 and E_1 states differs significantly from that of the trapped CO molecule ($\Theta = 69^\circ \pm 6^\circ$). Their vibrational spectra also differ significantly. The E_0 and E_1 bands are located $\sim 180\text{--}205 \text{ cm}^{-1}$ lower in frequency than the B_1 band for the trapped CO molecule in the ground vibrational state (Fig. 3a, b). As will be shown below, it is virtually impossible that the E_0 and E_1 bands are due to transitions from higher excited vibrational levels of the trapped CO. For that to be the case, the trapped CO molecule would have to be in the seventh excited vibrational state ($v = 7$). This estimate was made taking into account the $0 \rightarrow 1$ transition frequency and the anharmonic shift of 27.5 cm^{-1} for the photolyzed CO trapped in the primary docking site.⁶³ To excite the trapped CO to the seventh ($v = 7$) excited vibrational level, an energy of about $14,400 \text{ cm}^{-1}$ should be deposited into the vibrational mode of this molecule. In the present experiment, the $^1\pi\pi^*(Q)$ state of the carbonmonoxy chains was directly excited at 543 nm ($18,420 \text{ cm}^{-1}$). The bond energy required to break the bond between Fe and CO, which can be approximated by the measured enthalpy change for binding to the gaseous ligand,^{39,68} is $\sim 23.1 \text{ kcal/mol}$ ($\sim 8,080 \text{ cm}^{-1}$) (averaged over α and β subunits).⁶⁹ Therefore, at our photoexcitation conditions, $\sim 40\%$ of the photon energy would be spent just on breaking the bond between the heme iron and CO. Hence, the remaining energy ($\sim 10,340 \text{ cm}^{-1}$) is not sufficient for the excitation of the trapped CO to the seventh ($v = 7$) excited vibrational level. Based on this conclusion and on the abovementioned difference in the CO orientation, it is very unlikely that the E_0 and E_1 bands would correspond to highly excited vibrational states of the photolyzed CO trapped in the primary docking site. Moreover, it is unlikely that the E_0 and E_1 bands correspond to the lowest vibrational states of the photolyzed CO trapped in any secondary docking sites within the protein matrix, since, in this case, a very substantial downshift in the vibrational frequency is required to bring it to the similar value as for the heme-bound CO and nothing like that has been ever reported before. Therefore, we rule out that the positive transient E_0 and E_1 bands can originate from the photodissociated CO molecule and assign the E_0 and E_1 bands to the vibrational transitions of the CO ligand bound to the photoexcited heme.”

And we have added additional reference in the text:

63. Nuernberger, P., Lee, K. F., Bonvalet, A., Vos, M. H. & Joffre, M. Multiply excited vibration of carbon monoxide in the primary docking site of hemoglobin following photolysis from the heme. *J. Phys. Chem. Lett.* **1**, 2077–2081 (2010).

In the “Introduction” section (Page 3) we have also added a statement mentioning the orientation of the photolyzed CO molecule in the primary docking site, and mentioned the established time scale for the translocation of CO from its binding site (Fig. 1c, d) to the primary docking site:

“In less than 1 ps after the photodissociation,¹⁵ the CO molecule translocates from its binding site (Fig. 1c, d) to a primary docking site located above the heme group within a few angstroms¹⁶ from the heme iron. This translocation of CO involves ligand translation as well as rotation. The primary docking site constrains CO to lie approximately parallel to the plane of the heme,⁸ an orientation approximately orthogonal to that of the heme-bound CO. In the α and β subunits within human Hb at ambient temperature, the CO molecule remains trapped in the primary docking site at most a few nanoseconds.¹⁶”

3) There is a complete lack of literature review from time-resolved mid-IR. Several studies in the past have tried to dissociate Fe-CO by multiple IR photon absorption of 1950 cm⁻¹ in the ground state. Even 6 photon absorption did not manage to dissociate in the ground state. In the excited triplet ³T₁, the bond is weaker, so how many quanta of vibrational energy is needed to break the triplet Fe-CO bond? This should be estimated.

Response: We are grateful to the reviewer for pointing to that omission in our paper. In the “Discussion” section (Page 16) we have added the review of the attempted studies of Fe–CO bond dissociation by multiple IR photon absorption of 1950 cm⁻¹ in the ground state. The text now reads:

“The above mentioned conclusion concerning the impossibility of the CO dissociation in the vibrationally excited states of the ground electronic state at our photoexcitation conditions are indirectly confirmed by experiments with direct mid-IR excitation of the CO stretch vibration in the ground electronic state of HbCO, when up to six excited CO vibrational levels were populated.⁴⁷ In those experiments, it was tested if the Fe–CO bond breaking could be caused by an energy flow from the initially excited high-frequency CO stretch mode into the low-frequency Fe–CO stretch mode (if the coupling between the CO and the FeC bonds was sufficiently strong). However, no marked CO photodissociation was observed indicating that fast dissipation of the vibrational energy from the Fe–CO mode prevents the population of highly excited vibrational levels of that mode.”

To attempt to dissociate CO from the heme with vibrational excitation one would have to excite the low-frequency (~500 cm⁻¹) Fe–CO vibration into the 16th vibrational sub-level (as the binding energy is about 8,080 cm⁻¹) which is challenging.

With respect to the Fe–CO bonding energy in the triplet excited state, we agree that would be rather useful piece of information in the context of our study. However, to the best of our knowledge, there is no data available in literature on this subject. We will look into the opportunities to perform the theoretical modelling in future.

4) How is the energy transferred to CO modes? The absorption occurs in the heme xy-plane, and then is transferred to the z direction (His-Fe-CO bonds). There is no direct vibrational coupling between heme and CO, so the energy needs to flow via Fe-CO to the CO modes. If CO is active, it means that Fe-CO modes have a lot of energy and thus dissociation should occur before. Perhaps, since this process is very fast, again, there are simply different excited unbound CO channels that dissipate energy to the protein.*

Response: We thank the reviewer for bringing up the discussion of the mechanism for energy transfer from the heme to the Fe–CO bond, indeed this was not clearly addressed in our manuscript however

such mechanism was covered in earlier works. In the “Discussion” section in the beginning of the second paragraph on Page 17, we have added the following text:

“Strong Jahn-Teller distortions in this state afford an efficient energy transfer from the porphyrin plane (x, y axes) to the Fe–CO axis (z axis), activating dissociative stretching vibrations, which can lead to CO dissociation.³⁵”

On the previous page we have already highlighted why the observed E_0 and E_1 bands are not likely to originate from the photolyzed CO vibrationally excited.

Additionally in the “Discussion” section (Page 15, bottom paragraph, and Page 16) we have also included the discussion of the ultrafast energy redistribution over different vibrational modes in the heme proteins after absorption of the visible light:

“In the heme proteins, ultrafast energy redistribution over different vibrational modes has been thoroughly studied.³¹ It was established that, after absorption of the visible light, energy is transferred promptly to a highly excited vibronic manifold by Frank-Condon coupling. From this manifold, the vibrational energy is transferred with different timescales into low- and high-frequency modes, prior to slow dissipation through the protein on the ~ 10 – 20 ps timescale. In the ground electronic state, vibrationally excited levels of high-frequency modes ($\nu > 1,000$ cm^{-1}) are populated in a few hundred femtoseconds, and relax in several picoseconds after the photoexcitation.³¹ In contrast, vibrationally excited levels of low-frequency modes are populated quasi-instantaneously and are thermally equilibrated within ~ 300 fs.³¹ Taking into account that molecules such as hemes contain about one hundred vibrational modes, it is unlikely that, at 2 ps after the photoexcitation (the earliest time delay considered in our data analysis), a large amount of excess energy (up to $\sim 40\%$ of the absorbed photon energy, $\sim 8,080$ cm^{-1}) required to break the Fe–CO bond in the ground electronic state is still accumulated in a single vibrational mode such as the dissociative Fe–CO stretching mode at ~ 500 cm^{-1} .¹² Therefore, it is unlikely that the observed slow (~ 15 ps) CO photodissociation can be caused by vibrationally excited states of the heme–CO complex in the ground electronic state. It is unlikely that both the E_0 and E_1 states are the vibrationally excited states of the ground electronic state.”

5) The calculations of Eaton and coworkers are not sufficient for the interpretation of an extra channel for dissociation. They were simply based on crystal field theory, and Kasha’s rule, meaning that dissociation could potentially occur from the lowest singlet and the lowest triplet states. The main dissociation channel in sub-picosecond is clearly from singlet, but not 1T_1 as described by Eaton et al. Recent theory showed that indeed, 10-20% of the population ends up in the lowest metal-centered, but it is not shown to be dissociative. A further mechanistic explanation using modern theoretical techniques of such a dissociation channel is currently lacking and needed for the correct interpretation of the observed data.

Response: We agree with the reviewer pointing to the over-simplified level of calculations (by modern standards) reported in the earlier work by Eaton and coworkers. Instead we now refer to the results of the high-dimensional quantum dynamics calculations, published in more recent work which referenced in our manuscript:

35. Falahati, K., Tamura, H., Burghardt, I. & Huix-Rotllant, M. Ultrafast carbon monoxide photolysis and heme spin-crossover in myoglobin via nonadiabatic quantum dynamics. *Nat. Commun.* **9**, 4502 (2018).

We have approached the authors of this publication in private communication and asked their consent to interpret their theoretical results in the way confirming our assumption “that the $^5\text{MLCT}/^5\text{MC}$ excited states could be those dissociative ones in which the CO photodissociation occurs with the ~ 15 ps time constant” (here we quote our original text, now on Page 18).

In the “Discussion” section we have removed the text describing in detail the calculations of Eaton and coworkers. The following text has been removed:

“Our model for successive population of the two dissociative states (Fig. 7a) is in line with the energy level diagram (Fig. 7b) proposed by Eaton and colleagues³¹. The first dissociative state HbCO^*_1 (Fig. 7a) corresponds to the singlet excited ligand field $^1\text{T}_1$ state of HbCO in the range of 16,000 to 17,850 cm^{-1} (Fig. 7b). This ligand field, metal-centered (MC) state involves the promotion of an electron from d_{xz} or d_{yz} into the antibonding d_{z^2} orbital^{65,66}. In turn, the second dissociative state HbCO^*_{II} corresponds to the ligand field $^5\text{T}_2$ and/or $^3\text{T}_1$ state of HbCO (Fig. 7b), which are populated from the $^1\text{T}_1$ state. In both the $^5\text{T}_2$ and $^3\text{T}_1$ states, the antibonding d_{z^2} orbital is populated as well. The $^3\text{T}_1$ state was estimated³¹ to be the lowest excited state of HbCO , the $^5\text{T}_2$ state being nearby. The latter was considered^{15,67} to be the most plausible parent state for the high-spin deoxy-Hb. The $^3\text{T}_1$ state was also considered^{15,67} to be dissociative with respect to CO because of spin-orbit coupling of this state with the nearby $^5\text{T}_2$ state.”

We have also removed the simplified energy level diagram (which was on Fig. 7b in the first version of our manuscript) for the electronic states predicted to be involved in the photodissociation of CO from HbCO (according to W. Eaton and colleagues³¹).

Reviewer #2 (Remarks to the Author):

The authors reported CO photodissociation from the long-lived excited state(s) of HbCO and discussed the mechanism of heme-CO photodissociation based on unique picosecond to millisecond time-resolved mid-IR data. They attributed the observed spectral changes to the two-step photodissociation of CO and claimed that this is the first experimental proof of the existence of several dissociative states of HbCO. The results of this study will likely be of great interest to the readership of this journal. The interpretation of the experimental data may need to be reconsidered. Therefore, I recommend that the authors make appropriate revisions to publish this manuscript.

The authors detected a 15 ps component in the IR absorption spectral change by careful measurements. The quality of the measured data is high. It is convincing that this 15 ps component is not an artifact.

Response: We are extremely grateful to the reviewer for their recognition of the importance of our study and the quality of our data.

However, the conclusion that the breaking of the chemical bond between ferrous heme iron and CO includes a slower (~15 ps) process cannot be drawn from the experimental results reported in the manuscript by following reasons.

The authors ruled out the possibility that E_0 is the vibrationally excited state of the ground electronic state, as the vibrational excited states are typically shifted to a lower frequency relative to the fundamental frequency. However, in the model by Shelby, Harris, and Cornelius, the transition frequencies of the vibrationally excited states may shift to higher wavenumbers owing to the anharmonic coupling with other vibrational modes.

R. M. Shelby, C. B. Harris, P. A. Cornelius; The origin of vibrational dephasing of polyatomic molecules in condensed phases. J. Chem. Phys. 1 January 1979; 70 (1): 34–41.

Therefore, it is possible that CO stretching vibrations exhibit higher wavenumber even in the vibrationally excited state of the electronic ground state.

Response: We would like to thank the reviewer for pointing to that earlier work and for the suggestions how this might affect our interpretation of the experimental data. Indeed as correctly pointed by the reviewer, Shelby, Harris, and Cornelius predicted in their model⁷¹ that the vibrational frequencies for polyatomic molecules may shift to higher wavenumbers owing to the anharmonic coupling with other vibrational modes. However as is always the case with the vibrationally hot state, formation of vibrationally hot states must be matched with the corresponding depletion of the vibrational ground state absorption (e.g. ground state bleach). And consequently the decay of the hot vibrational state is accompanied by the matching recovery of the ground state bleach, occurring with the same time constant. As we stress repeatedly throughout our paper, we do not observe any sign of the ground state bleach recovery from the earliest recorded time delays up to nanosecond time scale, when the geminate ligand rebinding begins. Therefore we believe vibrationally hot state cannot account for the observed E_0 and E_1 transient bands, and our assignment still holds.

To reflect that additional discussion inspired by the reviewer, we have modified the text on Page 16 as follows:

“The fact that the E_0 and E_1 bands (Fig. 5a and 6a, insets) appear several wavenumbers higher in frequency than the heme-bound CO stretch bands (the 0→1 and 1→2 vibrational transitions, respectively) for the carbonmonoxy heme proteins in the ground electronic state rules out the assignment of the E_0 and E_1 bands to the vibrationally excited CO in the heme-CO complex in the ground electronic state. Shelby, Harris, and Cornelius predicted in their model⁷¹ that the vibrational frequencies for polyatomic molecules may shift to higher wavenumbers owing to the anharmonic coupling with other vibrational modes. However such anharmonic coupling would inevitably result in the corresponding bleach of the vibrational ground state (as typically observed in 2DIR experiments) which has to recover on the same timescale as the decay of the E_0 and E_1 bands, which we do not observe in the present case.”

The authors observed that a small increase (up to 20 %) of the total integrated area occurring with a time constant of ~20 ps (Supplementary Fig. 4c, d) and attributed it to the slow dissociation of CO. However, such an intensity increase can be observed for the vibrational relaxation of a dissociated CO molecule if the absorption cross-section of CO depends on the vibrational quantum number. Therefore, both experimental results can be explained by the vibrational relaxation of CO.

Response: We are thankful to the reviewer for such important comment. It is known from earlier works [64] and our data confirms (see Fig. 3) that the vibrational excitation of dissociated CO trapped within the protein matrix persists on the time scale of hundreds of picoseconds which is much longer than the observed grow-in of the vibrational signal of dissociated trapped CO (Fig. 3). Therefore, that grow-in process cannot be accounted for by the relaxation of the vibrationally excited free CO. We actually do observe the spectral signatures of the vibrationally excited dissociated CO (marked as B₁* and B₂* on Fig. 3a & 3b) and they indeed persist on the time scale much longer than 20 ps.

The following text has been added in the “Introduction” section (at the bottom of Page 4):

“The CO stretch bands for the heme-bound CO are susceptible to changes in the iron-ligand bond and the electric field at the active site.⁴⁶”

We have also added the following statement to the “Introduction” section (on Page 5):

“Upon photolysis, the vibrational bands reveal changes due to ligand relocation to docking sites within the protein, rotational motions of the ligand in these sites, and protein conformational changes,⁴⁶ time-resolved spectra of photolyzed CO being mainly determined by the interaction between the CO dipole and the surrounding electric field.”

And we have added an extra reference:

46. Nienhaus, K. & Nienhaus G. U. Ligand dynamics in heme proteins observed by Fourier transform infrared spectroscopy at cryogenic temperatures. *Methods Enzymol.* 437, 347–378 (2008).

Also the following paragraph has been added to the “Results” section (Page 8):

“The time-resolved vibrational spectra in Fig. 3a, b change noticeably with time. In particular, the total integrated area of all the transient bands corresponding to the photolyzed CO, produced in both the ground and first excited vibrational states, was found to increase up to 20 % with a time constant of ~20 ps (Fig. 3c, d). It should be noted that the observed increase in the total integrated area cannot be due to the vibrational relaxation of the photolyzed CO trapped inside the protein matrix, since the last process is much slower (hundreds of picoseconds)⁶⁴. The observed changes in the vibrational spectra can be induced by the actual increase in the population of the photodissociated CO on the ~20 ps timescale.”

If the authors' interpretation is correct, an electronic absorption band originating from the heme of HbCO(II) should appear after photoexcitation, which should decay at ~15 ps. Is such an electronic absorption band not observed?*

Response: We agree with the reviewer that it is quite tempting to try and identify the excited electronic state HbCO*_{II} in the visible transient absorption experiment, however it is a very challenging task to distinguish the HbCO*_{II} state signal due to very complex and highly convoluted nature of the transient absorption signal in the visible spectral range. There were numerous reports of transient absorption experiments for heme proteins in the visible spectral range, and here are just a few to mention:

Kholodenko, Y., Volk, M., Gooding, E. & Hochstrasser, R. M. Energy dissipation and relaxation processes in deoxy myoglobin after photoexcitation in the Soret region. *Chem. Phys.* **259**, 71–87 (2000).

Franzen, S., Kiger, L., Poyart, C. & Martin, J.-L. Heme photolysis occurs by ultrafast excited state metal-to-ring charge transfer. *Biophys. J.* **80**, 2372–2385 (2001).

Ye, X. et al. Investigations of heme protein absorption line shapes, vibrational relaxation, and resonance Raman scattering on ultrafast time scales. *J. Phys. Chem. A* **107**, 8156–8165 (2003).

In addition to the 15 ps dissociative excited state discussed in the present work, there are several other processes contributing to the visible transient absorption signal on similar time scale. Those include the dominant photoinduced transient absorption spectra of the deoxy-like chains with the heme in the ground electronic state, the cooling of the vibrationally hot ground electronic state of the heme, and the protein relaxation (which induces the changes in the electronic absorption of the heme) following the main (ultrafast) photodissociation step. That is why we have chosen the mid-IR time-resolved technique probing in the 1900–2150 cm^{-1} spectral window as it is completely free from any signals by the heme or protein itself.

Reviewer #3 (Remarks to the Author):

Direct observation of two-channel photodissociation of carbon monoxide from the hemoglobin subunits

Context: The present work is focused on the photodissociation of the diatomic CO (carbon monoxide) from the heme of isolated subunits of hemoglobin. The authors employed pump-probe infra-red absorption spectroscopy in the 1 ps to 1 ms time range to measure the evolving population of CO photodissociated from the heme. The IR probing frequency is that of heme-liganded CO (1950 cm^{-1}) and dissociated CO ($2125 - 2135\text{ cm}^{-1}$) and we may consider the phenomenon to be probed in the «ligand point of view».

Results: In the ps time range the populations of liganded CO and dissociated CO both show an evolution with a 15-ps time constant which is assigned to a second dissociation channel (11%) in addition to the main fast one ($\leq 50\text{ fs}$, 89%), thus from two distinct electronic excited states of the CO-liganded heme. In the slower time range, the kinetics of geminate (ns- μ s) and bimolecular (ms) rebinding of CO are well resolved. They appeared different for the α and β isolated subunits of Hb.

Rationale: The existence of the second slower photodissociation channel (15 ps) is based on the evolution of the CO populations. From its existence and previous studies, a model for electronic pathways of heme excited states leading to ligand dissociation is drawn.

The photodissociation of diatomics from the heme is indeed not fully explained and the existence of two electronic pathways is an important information. The data presented here are of good quality and rigorously analyzed. They support the conclusion and provide a proof of the existence of two photodissociation channels. I recommend publication of the present manuscript.

Response: We are very thankful to the reviewer for their high recognition of our work.

I do not have concerns, only a few comments and questions:

The authors wrote at the end of introduction (page 5 and also in page 12) that Eaton et al. (Ref. 31) have «first predicted the existence of several dissociative states of HbCO.» However, whereas Eaton et al. hypothesized which excited state could lead to dissociation (several are possible), they did not mean that several channels are simultaneously used. Also, Eaton et al. estimated an «upper limit for the dissociative time constant of 16 ps» for HbCO. We must note that this value was only based on the calculation of quantum yield (not on kinetics) which was very approximate since they used a high energy 8-ps excitation pulse at 353 nm (28328 cm^{-1}) for dissociating HbCO. Thus the value of 16 ps given by Eaton et al. should not be considered as a prediction of that measured in the present work.

Response: We are very grateful to the reviewer for this suggestion, which we are readily taking into account. In the last sentence on Page 5 (the initial version of the article), the following phrase was deleted: “as first predicted by Eaton and colleagues³¹”. Moreover, in the last paragraph, 2nd sentence on Page 12 (the initial version of the article), the following phrase was also deleted: “the theoretical prediction, made by Eaton and colleagues³¹, about”.

I suggest to the authors to incorporate Supplementary Figure 4 in the main manuscript (at least panels a and b) because these data strengthen those given in Fig. 2 and are necessary. Indeed, the increase of the induced absorption is a strong proof and cannot be dissociated from the evolution of the bleaching.

Response: We are very glad that the reviewer is very supportive in considering this data as the strong proof of the second (slower) channel of CO photodissociation. Indeed we considered originally to have that figure in the main text from start, however given the paper size limitation we ended up with moving this figure to the Supplementary. However now in the revised manuscript that figure is incorporated into the main text and now mentioned as Fig. 3. As a result, the necessary changes were made to the text. In the “Results” section (Pages 7 and 8) we have added the following text:

“The time-resolved vibrational spectra for CO photolyzed from the isolated carbonmonoxy α and β chains are shown in Fig. 3a and 3b, respectively. The photolyzed CO absorbs at frequencies between $2,090$ and $2,150\text{ cm}^{-1}$. Two major positive transient bands labeled B₁ and B₂ (Fig. 3a, b) correspond to the photolyzed CO molecules produced in the ground vibrational state ($v = 0$) and oriented oppositely in the primary docking site.^{8,62} For the α chains, the B₁ and B₂ bands are at $2,136$ and $2,124\text{ cm}^{-1}$,

while for the β chains those are at 2,133 and 2,123 cm^{-1} (Fig. 3a, b), respectively. The splitting between the maxima of the B_1 and B_2 bands observed for the α and β chains was found to be ~ 12 and ~ 10 cm^{-1} , respectively, agreeing well with that of 10.5 cm^{-1} published for HbCO.⁶³ Two satellite features labeled B_1^* and B_2^* (Fig. 3a, b) arise from the CO generated in the first excited vibrational state ($v = 1$).⁶⁴ (In Fig. 3a, the B_2^* band is not well resolved due to signal to noise limitation.) The B_1^* and B_2^* bands is a copy of two major B_1 and B_2 bands with lower amplitude and red shifted due to the anharmonicity of the vibrational ladder of CO.⁶³ The population of CO in its first excited vibrational state ($v = 1$) was found to be about 5%. This value was obtained within the harmonic oscillator approximation, where the integrated absorbance for a vibrational transition is proportional to $(v + 1)$.⁶⁵ Here, the absorbance cross section for the first hot band transition ($1 \rightarrow 2$) was assumed to be twice of that for the ground state transition ($0 \rightarrow 1$). Therefore, under our experimental conditions, photolysis of the heme proteins produces a small population of CO in its first excited vibrational state. Moreover, the photolyzed CO is not being formed in any vibrational states higher than $v = 1$. The decay of the first excited vibrational state ($v = 1$) of the photolyzed CO trapped in the primary docking site is known to occur over hundreds of picosecond.⁶⁴

The initial anisotropy of the vibrational spectra for photolyzed CO was used to determine the orientation of unbound CO located in the primary docking site (for details see Supplementary Note 2). Assuming that the IR transition dipole of the CO molecule is along the CO axis, the average angle Θ between the CO bond and the heme plane normal in both Hb chains was found to be $69^\circ \pm 6^\circ$ for both B_1 and B_2 photoproduct states. The obtained results indicate that, in the primary docking site of both the α and β chains, CO lies approximately in the plane of the heme (as opposed to the heme-bound CO.)

The time-resolved vibrational spectra in Fig. 3a, b change noticeably with time. In particular, the total integrated area of all the transient bands corresponding to the photolyzed CO, produced in both the ground and first excited vibrational states, was found to increase up to 20 % with a time constant of ~ 20 ps (Fig. 3c, d). It should be noted that the observed increase in the total integrated area cannot be due to the vibrational relaxation of the photolyzed CO trapped inside the protein matrix, since the last process is much slower (hundreds of picoseconds)⁶⁴. The observed changes in the vibrational spectra can be induced by the actual increase in the population of the photodissociated CO on the ~ 20 ps timescale.”

In the “Results” section (Page 10, bottom paragraph) we have added the following two statements:

“It should be noted that the ground state bleach signal due to the ligand dissociation is spectrally separated from the photolyzed CO absorption signal by about 180 cm^{-1} and differ by more than an order of magnitude in absorbance. For the isolated carbonmonoxy α and β chains, the integrated area of the bleach signal was found to be 36 ± 3 times larger than that of the photolyzed CO absorption signal at the magic angle polarization setting at a delay of 100 ps, agreeing well with the data for tetrameric Hb¹³CO.⁶²”

*In Fig. 5a the fitted kinetics do not strictly follow the experimental points. Is this due to the fact that the authors imposed exactly the same time constant for fitting the three polarized kinetics? Contrarily, the fitted kinetics in the Supplementary Figures go through the experimental points.
Can the authors amend the figure or comment?*

Response: We are thankful to the reviewer for pointing to this fact. This originated from the fact that for the results shown in Supplementary Figure 3 the data for different polarisations were fitted independently of each other and this resulted in better match between the fit and the experimental data points. However, the results plotted on Fig. 5a and 6a (Fig. 4a and 5a in the initial version of the article), the data obtained for different polarisations were globally fitted with the lifetime distribution $g(\log\tau)$ being the same at all these polarization settings. And exactly as noted by the reviewer, this resulted in increased deviation between the fit and the experimental data points. To clarify that, we have added the following text to the caption of Fig. 5a:

“Global SVD in (a), as in (b), was performed on the global data matrix composed of data obtained at three polarization settings. In (a), as in (b), the time-dependent amplitudes (V_1) of the first basis spectrum, multiplied by the first singular value (s_1), at three polarization settings were globally fitted to eqn (S5) with the lifetime distribution $g(\log\tau)$ being the same at all these polarization settings, the initial anisotropy r_0 (eqn (S6)) being the global parameter (see Supplementary Note 2 for details). Additionally, at the global fitting in (b), the rotational correlation time τ_{rot} (eqn (S6)) was taken as a global parameter.”

And we have also added the following text to the caption to Fig. 6a:

“Global SVD in (a), as in (b)–(c), was performed on the global data matrix composed of data obtained at three polarization settings. Description of (a) and (b) is the same as for Fig. 5.”

The authors calculated a 400-ps time constant for the transition 1T_1 to 3T_1 states. It should be detectable by transient electronic absorption. For example in Ref. 26 a 300-ps component was reported, although assigned differently as here. Can the authors shortly comment on the assignment in Ref. 26?

Response: We are grateful to the reviewer for this very valuable comment. Indeed, Martin and co-workers interpreted the 300 fs process as the deactivation of excited electronic states to the ground state in their earlier work (Ref. 30 in our paper). However, in their follow-up publication (Ref. 12 in our paper), Martin and co-workers revised their assignment of that 300 fs process and re-assign it to the transition between electronic excited states.

In our revised paper we have added the following text in the beginning of the second paragraph on Page 15:

“A similar ~ 300 fs value (as $\tau_{\text{tr}} \sim 400$ fs obtained here) was reported earlier for a relaxation process in the heme proteins after the photoexcitation by the visible light^{12,30} and eventually assigned to the transition between different excited electronic states of the heme.¹²”

The spectral SVD component U_2 represents the small shift of the absorption band E_0 in the slow dissociation path with respect to the fast one. It is also discernable on Fig. 2b and e. I agree with the authors' justification to discard an electronic ground state origin. How to insert this information within the frame of the model of Fig. 7?

Response: We are very glad that the reviewer is fully supportive of our vision that the observed 15 ps process cannot originate from the ground electronic state of the heme. The second SVD basis spectrum U_2 (Fig. 6c, which was Fig. 5c in the initial version of the article) is determined by spectral changes caused by the interconversion between the A_0 and A_1 conformational substates after the CO photodissociation. We shall reiterate that the basis spectrum U_2 has only a small contribution into the overall signal, $\sim 1\%$.

We have addressed the reviewer's suggestion by adding the following clarification to the caption to Fig. 8 (which was Fig. 7 in the initial version of the article):

“To simplify the diagram and focus only on the CO photodissociation, protein conformational transitions are not shown. The diagram is labeled for the case of Hb, but it is generally applicable for each Hb subunit in a chosen conformational substate. For the α and β chains in the A_1 conformational substate, the lifetime distributions $g(\tau)$ are shown in Fig. 7 (black and blue line, respectively). The A_1 conformational substate is characterized by the infrared absorption band of the bound CO, $\nu_{\text{CO}}(A_1) \approx 1951 \text{ cm}^{-1}$; the E_0 state, $\nu_{\text{CO}}(E_0) \approx 1954 \text{ cm}^{-1}$.”

Additionally, there is some clarification to the caption to Fig. 6 (which was Fig. 5 in the initial version of the article):

“In (c), the time-dependent amplitude (V_2) of the second basis spectrum, multiplied by the corresponding singular value (s_2) is plotted at three polarization settings. The time course of s_2V_2 at the magic angle polarization setting is determined by (i) an overall decrease in amplitude due to the CO rebinding to the deoxygenated chains and (ii) spectral changes caused by the interconversion between

the A₀ and A₁ conformational substates. In the inset to (c), the second basis spectrum (U₂) and its fit with a Gaussian function are shown in black as circles and the solid line, respectively. U₂ exhibits a band at 1,968 cm⁻¹ near the stretching band of CO bound to the isolated Hb chains in the A₀ substate.⁴⁵

Although the authors provide a Ref. (61) to describe the difference in the geminate CO rebinding rate between α and β chains, they did not give explanation. Since the data are new, it is worth to add at least a sentence for providing a structural explanation.

Response: We thank the reviewer for this suggestion and we totally agree with that. To address it, we have added the following explanations on Page 10 (Ref. 16 in our revised paper is Ref. 61 in the initial version of the article):

“The prompt geminate CO rebinding phase in the β chains was found to be faster than the one in the α chains (Supplementary Table 4).”

“As revealed in the earlier study¹⁶ by time-resolved Laue crystallography of photolyzed HbCO, there is a correlation between the rate constant for the CO rebinding from the primary docking site within the distal heme pocket (Fig. 1c, d) and the CO center-of-mass displacement during this process. The distance between the CO binding site and the primary CO docking site in the β subunits has been found to be ~0.25 Å smaller than that in the α subunits (~1.83 Å at 100 ps), and the CO rebinding from the primary docking site is faster in the β subunits, suggesting distal control of the CO rebinding dynamics.”

This completes our Response to the reviewers.

We hope very much that we have responded satisfactorily to all the questions and comments from all the reviewers.

On behalf of all the authors,
Yours sincerely

Igor Sazanovich